# Evaluating the clinical utility of large language models for hepatocellular carcinoma treatment recommendations: A nationwide retrospective registry study

Keungmo Yang, Jaejun Lee, Jeong Won Jang, Pil Soo Sung, Ji Won Han *

Division of Gastroenterology and Hepatology, Department of Internal Medicine, College of Medicine, The Catholic University of Korea, Seoul, Republic of Korea

* tmznjf@catholic.ac.kr

## Abstract

### Background

Hepatocellular carcinoma (HCC) management requires complex decision-making considering tumor burden, liver function, and patient's functional performance status. Large language models (LLMs) show promise in clinical applications, but their utility in HCC treatment recommendations remains unexplored. We evaluated the clinical relevance of LLM-generated treatment recommendations by comparing concordance with real-world physician decisions and survival outcomes.

### Methods and findings

We analyzed 13,614 treatment-naive HCC patients diagnosed between 2008 and 2020 in the Korean Primary Liver Cancer Registry. Treatment recommendations were generated using ChatGPT 4o, Gemini 2.0, and Claude 3.5 with standardized prompts referencing the American Association for the Study of Liver Diseases and the European Association for the Study of the Liver guidelines. Patients were classified as "matched" when LLM recommendations corresponded to actual treatments received. Overall survival (OS) was compared between matched and mismatched groups, stratified by the Barcelona Clinic Liver Cancer (BCLC) stage. Decision tree analysis identified factors influencing treatment selection patterns. Concordance rates between LLM recommendations and physician decisions were 31.1% (ChatGPT 4o), 32.7% (Gemini 2.0), and 26.8% (Claude 3.5). In BCLC-A patients, treatment concordance with LLM recommendations was associated with significantly improved survival (ChatGPT 4o HR: 0.743, 95% CI [0.665, 0.831], $P < 0.001$). Conversely, in BCLC-C patients, concordance was associated with worse survival outcomes (ChatGPT 4o HR: 1.650, 95% CI [1.523, 1.787], $P < 0.001$; Gemini 2.0 HR: 1.586, 95% CI [1.470, 1.711], $P < 0.001$; Claude 3.5 HR 1.483, 95% CI [1.366, 1.610],

**Data availability statement:** De-identified data from the Korean Primary Liver Cancer Registry are available from the registry's data access committee for researchers who meet the criteria for access to confidential data (https://livercancer.or.kr/eng/main.php). The registry data cannot be publicly shared due to institutional and national privacy regulations. No new analytical code was generated for this study and the standardized prompts used for the large language model inferences are provided in Table 1. All relevant data are available within the manuscript and Supporting information files.

**Funding:** This work was supported by the National Research Foundation of Korea (NRF) grant funded by the Korea government (Ministry of Science and ICT) (RS-2025-23525359 to J.W.H.) funded by the Ministry of Health & Welfare, Republic of Korea.

**Competing interests:** The authors have declared that no competing interests exist.

**Abbreviations :** AASLD, American Association for the Study of Liver Diseases; AI, artificial intelligence; BCLC, Barcelona Clinic Liver Cancer; CIs, confidence intervals; EASL, European Association for the Study of the Liver; HCC, hepatocellular carcinoma; HRs, hazard ratios; IPTW, inverse probability of treatment weighting; LLMs, large language models; OS, overall survival; SMDs, standardized mean differences.

$P < 0.001$). In BCLC-B, concordance showed only modest or nonsignificant associations with survival across models. Decision tree analysis revealed that physicians prioritized liver function parameters, while LLMs emphasized tumor characteristics. In early-stage HCC, physicians avoided curative treatments when hepatic reserve was limited, whereas in advanced-stage HCC, physicians preferred locoregional therapies in patients with preserved liver function despite guideline recommendations for systemic therapy. This study is limited by its retrospective design, reliance on registry data without imaging information, and focus on guideline-era treatments, warranting future prospective validation.

## Conclusions

Concordance between LLM-generated and physician treatment decisions was associated with improved survival in early-stage HCC, whereas this association was not observed in advanced-stage disease. While LLMs may serve as adjunctive tools for guideline-concordant decisions in straightforward scenarios, their recommendations may reflect limited contextual awareness in complex clinical situations requiring individualized care. LLM recommendations should be interpreted cautiously alongside clinical judgment.

## Author summary

### Why was this study done?

- Liver cancer (hepatocellular carcinoma [HCC]) is common worldwide, and choosing the right treatment can be difficult because it depends on both the cancer stage and how well the liver is functioning.

- Although international guidelines provide recommendations, real-world treatment often varies because doctors tailor decisions to each patient's situation.

- Large language models (LLMs) such as ChatGPT, Gemini, and Claude can summarize medical information, but it is not known whether their treatment advice would match what doctors actually do in practice.

### What did the researchers do and find?

- We studied more than 13,000 patients with newly diagnosed HCC in South Korea and compared treatments suggested by three LLMs with the treatments patients actually received.

- We found that when LLM recommendations aligned with actual treatments, early-stage (BCLC-A) patients experienced improved survival. In contrast, for advanced-stage (BCLC-C) patients, concordance with LLM advice was linked to worse survival.

- Physicians tended to prioritize liver function, while LLMs emphasized tumor characteristics, leading to stage-dependent discrepancies.

## What do these findings mean?

- LLMs may help support straightforward treatment decisions that closely follow clinical guidelines, especially in early-stage cancer.

- However, they are not yet reliable for complex cases where doctors must consider many individual factors beyond what guidelines capture.

- LLM-generated advice should be used cautiously, only as a supplemental tool, and always alongside professional medical judgment.

- Because this study was retrospective and relied on registry data without imaging inputs, its findings require confirmation in future prospective studies.

## Introduction

Hepatocellular carcinoma (HCC) is the most common primary liver cancer and a major cause of cancer-related mortality worldwide [1]. Clinical decision-making in HCC management is particularly complex, as it must account for a heterogeneous set of factors, including tumor burden, underlying liver function, and the patient's overall performance status [2,3]. To guide these decisions, international societies such as the American Association for the Study of Liver Diseases (AASLD) and the European Association for the Study of the Liver (EASL) have proposed algorithms centered on the Barcelona Clinic Liver Cancer (BCLC) staging system [4,5]. However, several studies have shown considerable variability in real-world treatment decisions even among patients classified within the same BCLC stage, reflecting challenges in strict adherence to guidelines and the influence of physician preferences or institutional practice patterns [6–9]. These inconsistencies highlight the inherent heterogeneity of HCC management, which often defies strict adherence to standardized treatment algorithms.

Recent advances in artificial intelligence (AI), particularly in large language models (LLMs) such as ChatGPT, Gemini, and Claude, have led to growing interest in their clinical applications [10]. These models have shown notable promise across a range of medical domains, including summarizing clinical knowledge, answering guideline-based questions, and simulating clinical scenarios in response to natural language prompts [11]. Applications have spanned specialties from oncology to general medicine, often demonstrating high concordance with published recommendations [12–14]. However, most studies to date have focused on diagnostic accuracy, or educational utility, rather than therapeutic planning. Although LLMs are now accessible to both physicians and non-experts, their reliability and clinical appropriateness in supporting treatment decisions remain unvalidated.

Several studies have investigated the application of LLMs in liver diseases, including patient education, medical documentation, diagnostic assistance, and preliminary risk assessment [15]. In metabolic dysfunction-associated steatotic liver disease, ChatGPT has demonstrated high comprehensibility and moderate accuracy in providing lifestyle and management guidance, and has also shown diagnostic potential comparable to established non-invasive indices for disease screening [16]. Recent studies suggest that ChatGPT can also generate informative and partially accurate responses to frequently asked patient questions in the context of liver cirrhosis and HCC [17–19]. Despite previous evidence supporting the use of LLMs in patient education and information delivery, their clinical utility in supporting expert-level decision-making—particularly in real-world treatment selection and outcome prediction—remains underexplored.

PLOS Medicine

Therefore, the present study aimed to evaluate the clinical utility of LLM-generated treatment recommendations in HCC by comparing their concordance with real-world physician decisions and assessing their association with patient survival. Using a large, nationwide cohort of treatment-naive HCC patients, we also examined the stage-specific implications of this concordance and analyzed the underlying prioritization patterns of LLMs and physicians. Through this, we sought to determine whether LLMs can provide clinically relevant guidance in HCC treatment and identify factors influencing their effectiveness across stages.

## Methods

### Study patients

This study analyzed data from the Korean Primary Liver Cancer Registry, a nationwide, hospital-based cohort developed through collaboration between the Korean Liver Cancer Association and the Korea Central Cancer Registry using systematic sampling methods [20]. The registry contains comprehensive clinical information on newly diagnosed HCC patients across multiple hospitals in Korea, including data on tumor characteristics, liver function, underlying liver disease, treatment modality, and survival status. Ethical approval for this study was obtained from the Institutional Review Board of the College of Medicine, The Catholic University of Korea (KC25ZISI0058). A total of 19,774 treatment-naive patients diagnosed between 2008 and 2020 were included in the present analysis, and those with insufficient follow-up ($n=2,640$), missing laboratory data ($n=2,842$), unavailable imaging ($n=26$), unreported ECOG status ($n=168$), or undocumented treatment history ($n=484$) were excluded. The final study population comprised 13,614 treatment-naive patients (Fig 1). All patients

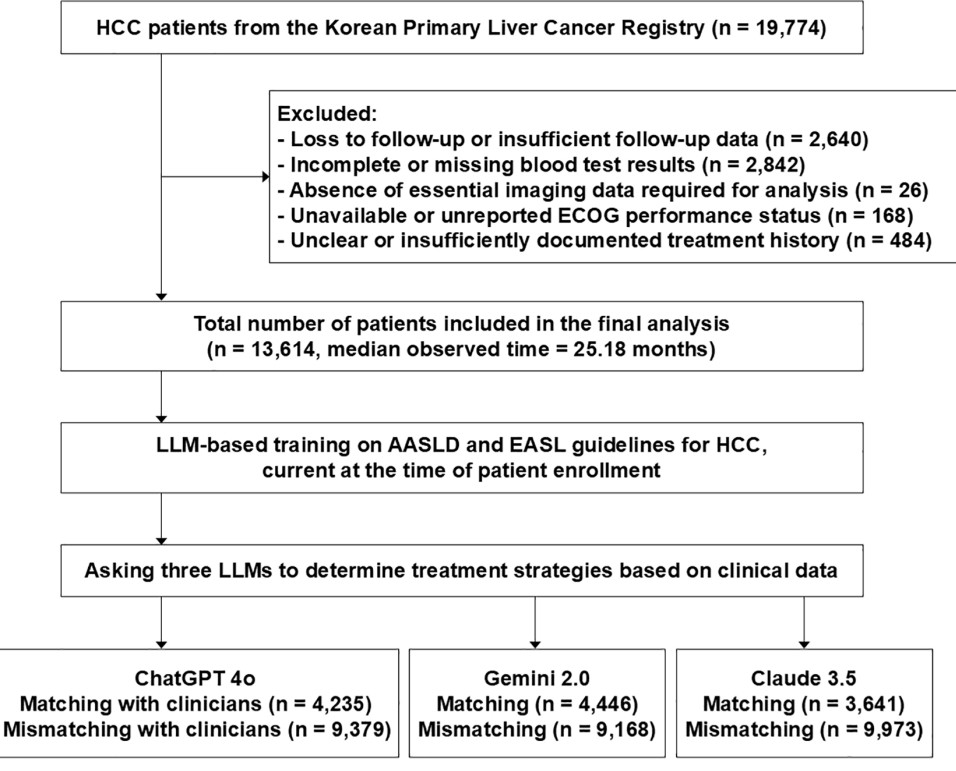

**Fig 1. Flowchart of patient selection and large language model-based treatment strategy comparison.** HCC, Hepatocellular carcinoma; ECOG, Eastern Cooperative Oncology Group; LLM, Large language model; AASLD, American Association for the Study of Liver Diseases; EASL, European Association for the Study of the Liver.

were staged at baseline according to the BCLC classification system, and detailed demographic, tumor-related, and laboratory data at the time of diagnosis were collected for analysis. Treatment data in the present cohort reflect therapies that were actually performed and documented in medical records, rather than physician recommendations that were not implemented. This study was conducted in accordance with the CHART guideline (S1 Table).

## Generation and evaluation of LLM-based treatment recommendations

Treatment recommendations were generated using three LLMs—ChatGPT 4o (OpenAI), Gemini 2.0 (Google), and Claude 3.5 (Anthropic)—by providing each model with identical clinical input variables alongside standardized prompts explicitly referencing the AASLD and EASL guidelines that were current at the time of each patient's diagnosis ensuring temporal consistency between the registry data and the reference standards used for training and evaluation [4,5]. All models were proprietary LLMs provided by their respective developers and used in their standard configuration without any fine-tuning or modification. While these LLMs were not fine-tuned on the guidelines themselves, they were queried using instruction-based prompts designed to elicit responses in alignment with established clinical protocols. All standardized prompts were derived directly from AASLD/EASL guidelines and were drafted by two board-certified hepatologists and one clinical researcher with expertise in HCC. Each model was independently tasked with selecting a treatment strategy based on the provided information. Detailed radiologic parameters such as segmental tumor location, vascular proximity, or anatomical feasibility were not available in the nationwide registry dataset. Therefore, the LLMs were provided solely with structured, guideline-level clinical variables (e.g., tumor size and number, vascular invasion, extrahepatic spread, Child–Pugh class, and ECOG performance status) that are routinely recorded in the registry.

Patients were then classified as "matched" if the LLM-generated recommendation corresponded to the treatment received in real-world practice, and "mismatched" otherwise. Two hepatologists independently assessed whether the LLM-recommended treatment corresponded to the actual therapy administered in clinical practice. Matched cases were defined based on clinically equivalent treatment categories rather than exact textual agreement. Variations in drug type, dosage, or duration were not regarded as discordant because such details were unavailable in the registry dataset. All three LLMs were queried using identical, guideline-based standardized prompts, which are now provided in Table 1. Each model inference was conducted twice to confirm reproducibility, using identical prompts and fixed decoding parameters. The ChatGPT 4o, Gemini 2.0, and Claude 3.5 models were accessed between March 1–5, 2025, using their publicly released versions available at that time. This classification enabled a comparative analysis of concordance rates across models and an evaluation of whether alignment with LLM recommendations was associated with differences in overall survival (OS). No patient received treatment based on LLM recommendations, and the raw outputs of LLM recommendations are provided in S1 Data. All therapies followed real-world physician decisions recorded in the registry, and LLM outputs were retrospectively compared with these treatments to assess concordance and related survival outcomes.

## Statistical analysis

Clinical characteristics of the study population were summarized using means with standard deviations for continuous variables and frequencies with percentages for categorical variables. Time zero was defined as the date of treatment initiation. Standardized mean differences (SMDs) were calculated to assess covariate balance before and after weighting, with SMDs < 0.1 indicating adequate balance. To evaluate the impact of LLM-generated recommendations, survival analyses were conducted using Kaplan–Meier estimates and log-rank tests. Cox proportional hazards regression models were used to estimate hazard ratios (HRs) and 95% confidence intervals (CIs) for OS, stratified by BCLC stage. Multivariate Cox models included covariates selected for their clinical relevance and statistical significance in univariate analyses.

To mitigate baseline imbalances among LLM recommendation groups, inverse probability of treatment weighting (IPTW) was applied using multinomial logistic regression to estimate generalized propensity scores. Stabilized and trimmed weights (1st to 99th percentile) were used to minimize extreme weight influence. Weighted analyses included

**Table 1. Standardized prompts for LLM-generated treatment recommendations.**

| **1. Primary instruction.** | |
| --- | --- |
| Input | "Could you help me decide on the best treatment option for patients with hepatocellular carcinoma (HCC) based on their clinical information and standard guidelines?" |
| **2. Patient-level dataset.** | |
| Input | "Review the attached CSV file containing clinical data of patients with hepatocellular carcinoma. Learn the structure and contents of this dataset thoroughly." |
| **3. Variable definitions.** | |
| Input | "Study the attached data dictionary, which describes each column in the dataset, including variable definitions, units, and valid ranges." |
| **4. Guideline learning according to the treatment year of the included patients.** | |
| Input | "Study the attached hepatocellular carcinoma treatment guidelines (PDF file). Internalize their algorithms and criteria sufficiently to make clinical decisions as a treating physician." |
| **5. Clinical reasoning task.** | |
| Input | "Assume the role of the attending physician evaluating each patient for the first time. Integrate all available clinical information—patient demographics, laboratory tests, liver function, imaging findings, vascular invasion, extrahepatic spread, and ECOG performance status." |
| **6. Guideline-concordant treatment selection.** | |
| Input | "Based on the guidelines and patient-specific variables, determine the single most appropriate treatment for each patient. Reflect real-world treatment availability in Korea when selecting systemic therapies: Lenvatinib available from 2018, and Atezolizumab + Bevacizumab available from 2020 onward." |
| **7. Output format.** | |
| Input | "Record exactly one final treatment per patient in a new column labeled treatment (e.g., resection, ablation, TACE, systemic therapy specifying regimen, or best supportive care). Do not list multiple options; provide a single best choice." |

ECOG, Eastern Cooperative Oncology Group; Transarterial Chemoembolization.

adjusted Kaplan–Meier curves and Cox models for OS. Variable importance was assessed using a decision tree model trained on treatment decisions, with the relative contribution of each clinical variable visualized as a percentage of the most influential feature. To aid interpretability, representative pruned decision trees for clinicians and each LLM, constructed using identical clinical covariates and Gini impurity–based recursive partitioning. All statistical analyses were conducted using R (version 4.5.0, R Foundation, Vienna, Austria), and a $P$ value of less than 0.05 was considered to indicate statistical significance.

## Results

### Treatment concordance between LLM-physician

Baseline characteristics of the 13,614 study patients are presented in Table 2. The mean age at diagnosis was 62.7 ± 11.5 years, and 79.2% and 20.8% of patients were male and female, respectively. Hepatitis B virus infection was the predominant etiology of HCC (55.3%), and most patients had preserved liver function, with 82.6% classified as Child-Pugh class A. The maximum intrahepatic tumor diameter was median 4.7 cm, and 40.6% of patients had multiple intrahepatic tumors. Portal vein invasion and extrahepatic metastasis were present in 24.2% and 10.7% of patients. Regarding BCLC stage, 43.2% of patients were classified as BCLC-A and 31.5% as BCLC-C.

Concordance between real-world physician treatment decisions and LLM-generated recommendations varied across models, with match rates of 31.1% for ChatGPT 4o, 32.7% for Gemini 2.0, and 26.8% for Claude 3.5 (Tables 2, S2, and S3). When comparing characteristics between matched and mismatch groups, the mismatched group had significantly higher rates of portal vein invasion (28.6% versus 14.4%, $P < 0.001$) and extrahepatic metastasis (15.3% versus 8.6%,

**Table 2. Baseline clinical characteristics according to concordance between physician decisions and ChatGPT 4o-generated treatment recommendations.**

| Clinical characteristics | Overall ($n^1$ = 13,614) | Treatment concordance with ChatGPT | | P value[2] |
|---|---|---|---|---|
| | | Mismatch ($n^1$ = 9,379) | Match ($n^1$ = 4,235) | |
| **Age at diagnosis** | 62.66 ± 11.54 | 62.63 ± 11.67 | 62.72 ± 11.26 | 0.065 |
| **Sex** | | | | 0.399 |
| Male | 10,783 (79.2%) | 7,410 (79.0%) | 3,373 (79.6%) | |
| Female | 2,831 (20.8%) | 1,969 (21.0%) | 862 (20.4%) | |
| **Diabetes mellitus** | 3,998 (29.4%) | 2,757 (29.4%) | 1,241 (29.3%) | 0.919 |
| **Hypertension** | 5,212 (38.3%) | 3,583 (38.2%) | 1,629 (38.5%) | 0.775 |
| **Hepatitis B** | 7,526 (55.3%) | 5,118 (54.6%) | 2,408 (56.9%) | 0.013 |
| **Hepatitis C** | 1,651 (12.1%) | 1,118 (11.9%) | 533 (12.6%) | 0.281 |
| **Past smoking history** | 6,177 (45.4%) | 4,205 (44.8%) | 1,972 (46.6%) | 0.063 |
| **Past alcohol use** | 4,991 (36.7%) | 3,427 (36.5%) | 1,564 (36.9%) | 0.673 |
| **ECOG performance status** | | | | 0.003 |
| 0 | 6,773 (49.8%) | 4,595 (49.0%) | 2,178 (51.4%) | |
| 1 | 4,032 (29.6%) | 2,822 (30.1%) | 1,210 (28.6%) | |
| 2 | 2,442 (17.9%) | 1,731 (18.5%) | 711 (16.8%) | |
| 3 | 231 (1.7%) | 147 (1.6%) | 84 (2.0%) | |
| 4 | 136 (1.0%) | 84 (0.9%) | 52 (1.2%) | |
| **Albumin (g/dL)** | 3.72 ± 0.67 | 3.72 ± 0.67 | 3.73 ± 0.67 | 0.996 |
| **Total bilirubin (mg/dL)** | 1.62 ± 2.86 | 1.62 ± 2.81 | 1.64 ± 2.95 | 0.280 |
| **INR** | 1.16 ± 0.55 | 1.16 ± 0.64 | 1.15 ± 0.23 | 0.214 |
| **Creatinine (mg/dL)** | 0.98 ± 0.78 | 0.98 ± 0.77 | 0.98 ± 0.80 | 0.077 |
| **Sodium (mmol/L)** | 138.17 ± 5.03 | 138.20 ± 4.86 | 138.08 ± 5.38 | 0.128 |
| **ALT (IU/mL)** | 54.43 ± 100.68 | 53.97 ± 107.84 | 55.45 ± 82.64 | 0.291 |
| **Platelet ($10^3$/uL)** | 164.01 ± 93.49 | 164.17 ± 93.57 | 163.65 ± 93.32 | 0.361 |
| **AFP (ng/mL)** | 13,409.30 ± 104,384.94 | 14,210.86 ± 108,575.02 | 11,634.11 ± 94,434.07 | 0.493 |
| **Multiple tumors** | 5,532 (40.6%) | 3,433 (36.6%) | 2,099 (49.6%) | < 0.001 |
| **Maximum tumor diameter (cm)** | 4.70 ± 3.91 | 4.77 ± 3.96 | 4.56 ± 3.78 | < 0.001 |
| **Portal vein invasion** | 3,298 (24.2%) | 2,687 (28.6%) | 611 (14.4%) | < 0.001 |
| **Hepatic vein invasion** | 773 (5.7%) | 564 (6.0%) | 209 (4.9%) | 0.012 |
| **Bile duct invasion** | 366 (2.7%) | 271 (2.9%) | 95 (2.2%) | 0.034 |
| **Hepatic artery invasion** | 158 (1.2%) | 115 (1.2%) | 43 (1.0%) | 0.301 |
| **Lymph node metastasis** | 976 (7.2%) | 633 (6.7%) | 343 (8.1%) | 0.005 |
| **Extrahepatic metastasis** | 1,457 (10.7%) | 811 (8.6%) | 646 (15.3%) | < 0.001 |
| **Ascites** | | | | 0.051 |
| None | 10,098 (74.2%) | 6,930 (73.9%) | 3,168 (74.8%) | |
| Mild | 2,217 (16.3%) | 1,573 (16.8%) | 644 (15.2%) | |
| Moderate to severe | 1,299 (9.5%) | 876 (9.3%) | 423 (10.0%) | |
| **Hepatic encephalopathy grade** | | | | 0.423 |
| None | 13,277 (97.5%) | 9,136 (97.4%) | 4,141 (97.8%) | |
| Grade 1 or 2 | 266 (2.0%) | 193 (2.1%) | 73 (1.7%) | |
| Grade 3 or 4 | 71 (0.5%) | 50 (0.5%) | 21 (0.5%) | |
| **Child-Pugh classification** | | NA | NA | 0.192 |
| A | 11,240 (82.6%) | 7,748 (82.6%) | 3,492 (82.5%) | |
| B | 2,241 (16.5%) | 1,549 (16.5%) | 692 (16.3%) | |
| C | 133 (1.0%) | 82 (0.9%) | 51 (1.2%) | |

*(Continued)*

**Table 2.** (Continued)

| Clinical characteristics | Overall ($n^1$=13,614) | Treatment concordance with ChatGPT | | P value[2] |
|---|---|---|---|---|
| | | Mismatch ($n^1$ = 9,379) | Match (n¹ = 4,235) | |
| **BCLC stage** | | | | < 0.001 |
| A | 4,064 (29.9%) | 2,923 (31.2%) | 1,141 (26.9%) | |
| B | 5,265 (38.7%) | 3,186 (34.0%) | 2,079 (49.1%) | |
| C | 4,285 (31.5%) | 3,270 (34.9%) | 1,015 (24.0%) | |
| **MELD score** | 9.81±4.05 | 9.82±3.98 | 9.80±4.19 | 0.008 |

[1]n (%); Mean±SD,

[2]Fisher's exact test.

ECOG, Eastern Cooperative Oncology Group; INR, international normalized ratio; ALT, Alanine aminotransferase; AFP, alpha-fetoprotein; BCLC, Barcelona Clinic Liver Cancer; MELD, model for end-stage liver disease.

$P<0.001$) compared to those in the matched group. Additionally, the mismatched group exhibited a lower proportion of multiple tumors (36.6% versus 49.6%, $P<0.001$) despite having a larger mean maximum tumor diameter (4.77 cm versus 4.56 cm, $P<0.001$). With respect to cancer staging, BCLC-C was more frequently represented in the mismatched group (34.9% versus 24.0%), whereas BCLC-B was more common in the matched group (49.1% versus 34.0%) (Table 2).

## Impact of concordance between LLM-physician on OS

Next, OS was assessed based on the concordance between LLM-generated treatment recommendations and real-world treatment decisions, stratified by BCLC stage and LLM model. In the entire cohort, patients whose treatments matched ChatGPT 4o recommendations exhibited significantly better survival outcomes compared to mismatched cases (HR, 0.946; $P=0.011$; Fig 2A). This survival difference was most evident in BCLC-A, where matched cases demonstrated substantially longer survival (HR, 0.626; $P<0.001$; Fig 2B). In BCLC-B, survival was slightly worse in matched patients, although the difference was relatively modest (HR, 1.091; $P=0.008$; Fig 2C). In contrast, in BCLC-C, matched patients showed worse survival outcomes compared to mismatched cases (HR, 2.271; $P<0.001$; Fig 2D). Similar trends were observed for other LLM models, such as Gemini and Claude (S1 and S2 Figs), indicating that concordance was associated with better clinical outcomes primarily in early-stage HCC but not in advanced-stage HCC.

## Stage-specific associations between LLM-physician with OS

Univariate Cox regression analysis (S4 Table) demonstrated that treatment concordance with LLM-generated recommendations was significantly associated with OS across all BCLC stages. However, in multivariable analysis adjusting for relevant clinical covariates (Table 3), matched treatment remained independently associated with improved survival in BCLC-A, with HRs less than 1 across all models: ChatGPT 4o (HR, 0.743; 95% CI [0.665, 0.831]; $P<0.001$), Gemini 2.0 (HR, 0.902; 95% CI [0.819, 0.992]; $P=0.034$), and Claude 3.5 (HR, 0.849; 95% CI [0.767, 0.940]; $P=0.002$). Furthermore, in BCLC-C, matched treatment exhibited significantly worse survival, with adjusted HRs exceeding 1: ChatGPT 4o (HR, 1.650; 95% CI [1.523, 1.787]; $P<0.001$), Gemini 2.0 (HR, 1.586; 95% CI [1.470, 1.711]; $P<0.001$), and Claude 3.5 (HR, 1.483; 95% CI [1.366, 1.610]; $P<0.001$). However, in BCLC-B, the survival benefit of LLM-concordant treatments was not statistically significant for ChatGPT 4o (HR, 1.008; 95% CI [0.943, 1.078]; $P=0.810$) and Claude 3.5 (HR, 1.058; 95% CI [0.978, 1.104]; $P=0.053$), with only Gemini 2.0 showing a marginal association (HR, 1.150; 95% CI [1.075, 1.229]; $P<0.001$). In era-stratified analyses (2008–2012, 2013–2016, 2017–2020), HRs for OS were consistent across calendar periods within each BCLC stage (S5 Table), suggesting that secular trends did not influence the observed associations. These findings suggest that LLM concordance showed stage-dependent associations with OS—positive in early-stage and negative in advanced-stage HCC.

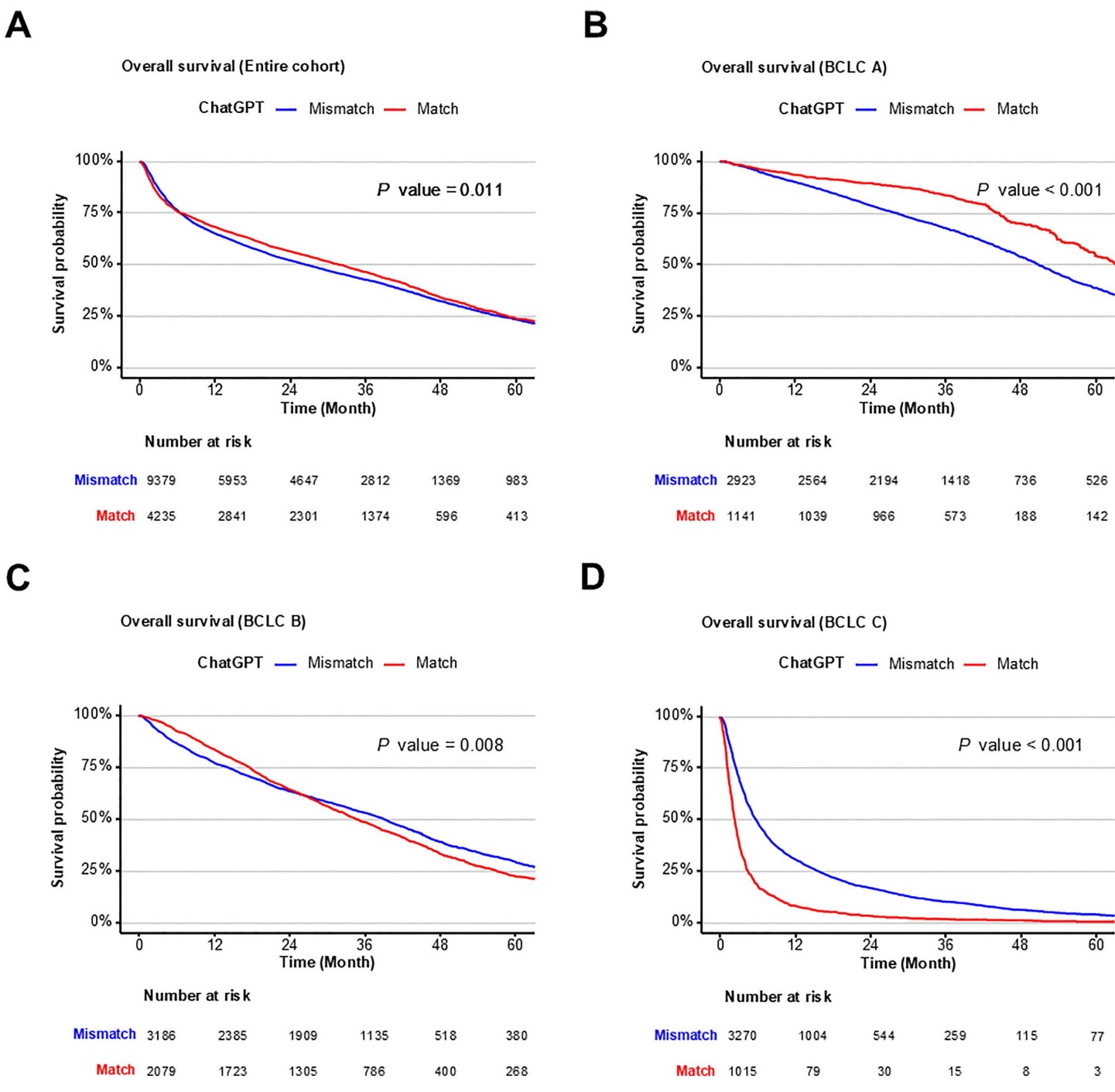

**Fig 2. Overall survival according to concordance between ChatGPT 4o-recommended and physician-administered treatments in HCC patients.** Kaplan–Meier survival curves comparing overall survival between patients whose treatment matched ChatGPT's recommendation (red line) and those whose treatment differed (blue line). **(A)** Entire cohort; **(B)** BCLC stage A; **(C)** BCLC stage B; **(D)** BCLC stage C.

**Table 3. Multivariate analyses of overall survival in HCC patients according to BCLC stage.**

| Clinical characteristics | BCLC stage A | | | BCLC stage B | | | BCLC stage C | | |
|---|---|---|---|---|---|---|---|---|---|
| | HR | 95% CI | *P* value | HR | 95% CI | *P* value | HR | 95% CI | *P* value |
| ChatGPT 4o-matched decision | 0.743 | 0.665, 0.831 | < 0.001 | 1.008 | 0.943, 1.078 | 0.810 | 1.650 | 1.523, 1.787 | < 0.001 |
| Gemini 2.0-matched decision | 0.902 | 0.819, 0.992 | 0.034 | 1.150 | 1.075, 1.229 | < 0.001 | 1.586 | 1.470, 1.711 | < 0.001 |
| Claude 3.5-matched decision | 0.849 | 0.767, 0.940 | 0.002 | 1.058 | 0.978, 1.104 | 0.053 | 1.483 | 1.366, 1.610 | < 0.001 |

HCC, hepatocellular carcinoma; BCLC, Barcelona Clinic Liver Cancer; HR, hazard ratio; CI, confidence interval. *P* values were calculated from multivariate Cox proportional hazards models.

## Additional validation analyses of model robustness

To address potential immortal-time bias, a 1-year landmark analysis was conducted, restricting the cohort to patients who survived at least 12 months after treatment initiation and initiating follow-up from the landmark time point. Even after this adjustment, the stage-dependent pattern of survival differences persisted (S3–S5 Figs). To further evaluate the robustness of causal associations, we additionally performed IPTW and doubly robust (augmented IPTW) Cox proportional hazards analyses for OS according to adherence to each LLM's treatment recommendation (S6 and S7 Tables). Across all models, covariate balance was adequately achieved after weighting, with absolute SMDs < 0.1 for all variables (S6 Fig). In IPTW-weighted analyses, adherence to LLM-generated recommendations in BCLC A was associated with significantly lower mortality risk—ChatGPT 4o (HR 0.847, *P* = 0.005), Gemini 2.0 (HR 0.869, *P* = 0.004), and Claude 3.5 (HR 0.846, *P* = 0.001)—whereas in BCLC C, matched cases exhibited higher risk—ChatGPT 4o (HR 1.513, *P* < 0.001), Gemini 2.0 (HR 1.518, *P* < 0.001), and Claude 3.5 (HR 1.464, *P* < 0.001) (S6 Table). Doubly robust analyses yielded consistent findings, with similar directional effects (S7 Table): ChatGPT 4o(HR 0.789, *P* < 0.001 for BCLC A; HR 1.571, *P* < 0.001 for BCLC C). These supplementary analyses confirm the stage-dependent association between LLM concordance and survival, indicating that the observed patterns remained robust across alternative causal modeling approaches.

## Factors underlying LLM-physician discordance

Decision tree models were constructed to assess key clinical variables influencing treatment selection in both physician decisions and LLM-generated recommendations (Fig 3). In the physician model (Fig 3A), BCLC stage was the most influential factor, followed by platelet count and INR. All three LLM models—ChatGPT 4o, Gemini 2.0, and Claude 3.5—also prioritized BCLC stage, but subsequently placed greater emphasis on tumor factors such as tumor size, portal vein invasion, and number of tumors, with relatively lower importance assigned to liver function parameters (Fig 3B–3D). Representative simplified decision trees for clinicians and each LLM are additionally presented in S7 Fig. These findings suggest that while both physicians and LLMs rely importantly on BCLC staging, physicians tend to consider liver function more prominently, whereas LLMs are more influenced by tumor characteristics.

Given the paradoxical survival outcomes observed in BCLC-A and BCLC-C, we performed subgroup analyses to investigate which clinical factors were associated with LLM-physician discordance. In BCLC-A, patients in the mismatched group were older (63.5 versus 60.1 years, *P* < 0.001), less frequently male (71.7% versus 76.2%, *P* = 0.004), and had poorer liver function, including lower albumin (3.78 versus 4.08 g/dL, *P* < 0.001), higher INR (1.17 versus 1.10, *P* < 0.001), and lower platelet counts (124.5 versus 153.4 × 10³/μL, *P* < 0.001) compared to the matched group (S8 Table). Thus, in BCLC-A, LLMs might recommend potentially curative treatments based on guidelines, whereas physicians might select other options when liver function was impaired.

Conversely, in BCLC-C, patients in the mismatched group had better liver function compared to the matched group, with higher albumin (3.54 versus 3.27 g/dL, *P* < 0.001), lower bilirubin (2.28 versus 3.47 mg/dL, *P* < 0.001), and lower INR (1.21 versus 1.27, *P* < 0.001), along with a significantly lower rate of extrahepatic metastasis (23.7% versus 62.4%,

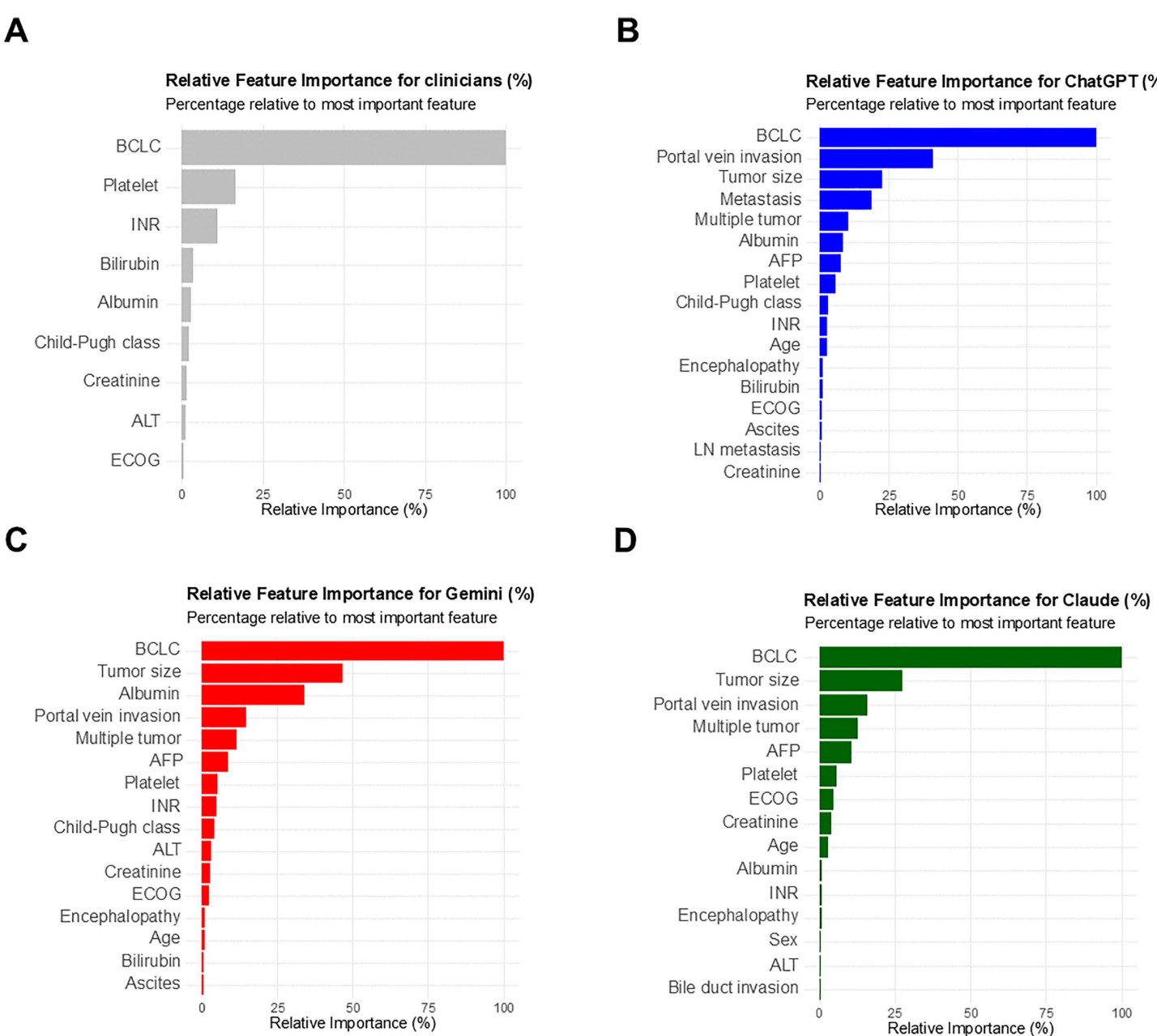

**Fig 3. Relative feature importance for treatment decision-making using a decision tree classifier. (A)** Feature importance from a decision tree model trained on physician decisions. **(B–D)** Feature importance derived from decision tree classifiers built using ChatGPT 4o (B), Gemini 2.0 (C), and Claude 3.5 (D) recommendations. Feature importance is presented as a percentage relative to the most influential variable (BCLC stage).

$P < 0.001$) (S9 Table). Thus, in BCLC-C, LLMs might suggest systemic or palliative care based on guideline-directed algorithms, but physicians might choose other options when hepatic reserve was preserved and extrahepatic spread was limited, reflecting a preference for locoregional control in select patients. Similar trends were observed with the Gemini and Claude models (S10–S13 Tables).

## Patterns of LLM-physician discordance

We further analyzed discordance patterns between LLM-generated treatment recommendations and real-world physician decisions. The frequency matrix showed that the greatest number of concordant cases with ChatGPT 4o and physicians occurred with conventional transarterial chemoembolization (TACE) (n = 2,079) and surgical resection (n = 964) (S8A Fig). ChatGPT 4o demonstrated the highest treatment concordance with conventional TACE (43.9%) and best supportive care (42.9%), whereas the lowest concordance was observed with lenvatinib (1.8%) and liver transplantation (9.7%) (S8B Fig). Similar trends were observed with the Gemini and Claude models, which showed greater concordance for guideline-based locoregional and potentially curative treatments and lower agreement in systemic therapies (S9 and S10 Figs).

Subgroup analyses by BCLC stage revealed stage-dependent discordance patterns between LLM recommendations and clinical practice (Figs 4, S11, and S12). Prevalence matrices of treatment recommendations across the three LLMs for BCLC stages A and C are presented in S13 Fig. In BCLC-A, LLMs frequently recommended potentially curative treatments such as surgical resection or RFA, whereas physicians often administered conventional TACE instead. In BCLC-C, systemic therapies or best supportive care were commonly suggested by LLMs, but conventional TACE was frequently used in real-world practice. The most frequent mismatch across the entire cohort was LLMs recommending conventional TACE when surgical resection had been performed (S14 Table). In stage-specific analyses, the most common discordance in BCLC-A involved recommendations for surgical resection in patients who received TACE or RFA, while in BCLC-C, LLMs most frequently recommended systemic treatment for patients who were treated with best supportive care or conventional TACE (Table 4).

To further clarify whether the observed survival differences reflected true concordance rather than treatment selection, we performed a within-treatment sensitivity analysis (S15 Table). Median OS for concordant cases did not uniformly exceed treatment-level baselines, being similar or even lower in several modalities (e.g., RFA in BCLC-A and best supportive care in BCLC-C). To better understand these discrepancies, we next analyzed how LLMs influenced therapeutic tier recommendations within each BCLC stage. As shown in S16 Table, LLMs more frequently proposed higher-tier, curative-intent therapies in early-stage disease (BCLC A), whereas in advanced stages (BCLC C) they more often suggested lower-tier or palliative options. In stage-specific IPTW analyses, higher-tier recommendations were consistently associated with improved OS (S17 Table).

## Comparative evaluation of survival outcomes across LLM models

Among 13,614 patients included in the final analysis, 6,078 (44.6%) received treatments concordant with at least one LLM recommendation (Fig 5). After random allocation of patients with overlapping matches, 2,190, 2,297, and 1,591 patients were assigned to the ChatGPT 4o, Gemini 2.0, and Claude 3.5 groups, respectively, for comparative survival analysis evaluating the efficacy of LLM-guided treatment recommendations (Fig 5). Before IPTW adjustment, baseline characteristics showed modest differences across the three groups (S18 Table). After IPTW application, covariate balance was substantially improved, with all SMDs reduced below 0.1 (S19 Table). Kaplan–Meier analysis of OS prior to IPTW adjustment showed no significant differences in survival between the three LLM-aligned groups in the entire cohort (P = 0.252; S14A Fig). However, subgroup analysis revealed that ChatGPT 4o-guided treatment was associated with a slight survival benefit in BCLC-A, with no significant differences in BCLC-B or C (S14B–S14D Fig). After IPTW adjustment, survival curves were closely aligned across models, and no statistically significant differences were observed in the entire cohort (P = 0.879; Fig 6A) or within BCLC stages A–C (P = 0.073 for A; 0.438 for B; 0.388 for C; Fig 6B–6D). These findings suggest that survival outcomes did not substantially differ among the three LLM-guided treatment groups.

## Discussion

This study assessed the clinical utility of treatment recommendations generated by LLMs (ChatGPT 4o, Gemini 2.0, and Claude 3.5) in patients with HCC using a large-scale, nationwide real-world cohort. Our findings demonstrated variable concordance rates (26.8%–32.7%) between actual clinical decisions and LLM-based recommendations, with significant

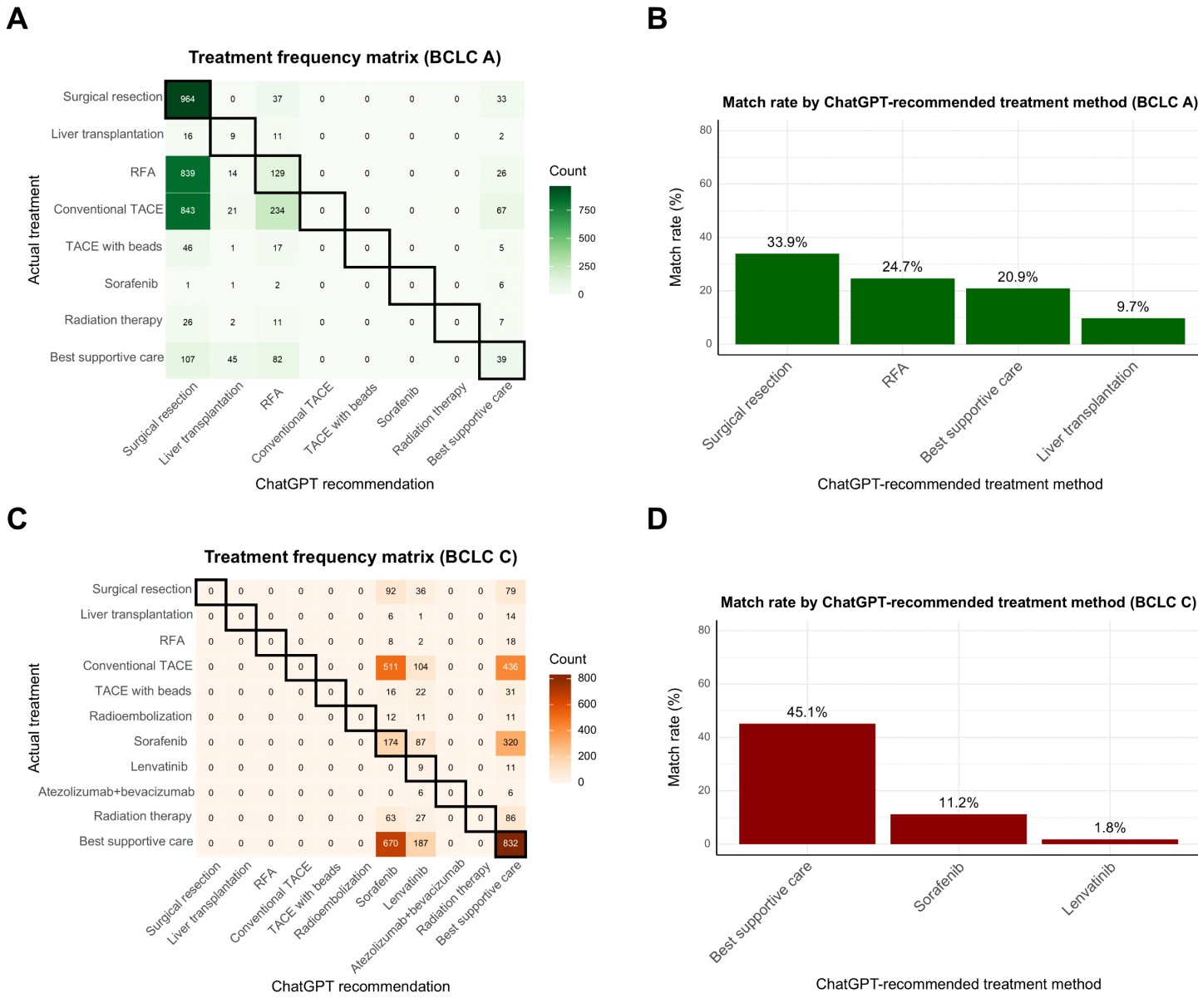

**Fig 4. Subgroup concordance analysis between ChatGPT 4o-based treatment suggestions and clinical practice across BCLC stages. (A)** Matrix visualization of treatment frequencies comparing ChatGPT-recommended options (x-axis) with actual therapies provided by clinicians (y-axis) in patients with BCLC stage A. Cells on the diagonal denote concordant decisions, with darker shades indicating higher agreement. **(B)** Bar plot summarizing the match rate (%) for each treatment recommended by ChatGPT among patients with BCLC stage A. **(C)** Frequency matrix for BCLC stage C patients, showing alignment between ChatGPT's suggestions and real-world physician decisions. **(D)** Match rates for each ChatGPT recommendation in BCLC stage C patients. To ensure interpretability, treatment categories with fewer than five cases were excluded from the bar plots.

survival benefits observed for concordant cases in BCLC-A but paradoxically worse outcomes in BCLC-C. Decision tree analysis revealed distinct prioritization patterns, highlighting physicians' greater emphasis on hepatic reserve versus LLMs' preference for tumor-related factors. Recent advancements in LLMs have prompted investigations into their applicability for clinical decision-making across various liver diseases, yet most prior studies were confined to assessing informational accuracy or educational utility [16–19,21,22]. To our knowledge, this is the first study to systematically evaluate

**Table 4. Top 10 most common disagreement patterns between LLM-generated recommendations and real-world treatments in patients with BCLC stage A and C.**

**ChatGPT 4o**

| Rank | BCLC stage A | | | BCLC stage C | | |
|---|---|---|---|---|---|---|
| | Actual treatment | ChatGPT recommendation | Count | Actual treatment | ChatGPT recommendation | Count |
| 1 | Conventional TACE | Surgical resection | 843 | Best supportive care | Sorafenib | 670 |
| 2 | RFA | Surgical resection | 839 | Conventional TACE | Sorafenib | 511 |
| 3 | Conventional TACE | RFA | 234 | Conventional TACE | Best supportive care | 436 |
| 4 | Best supportive care | Surgical resection | 107 | Sorafenib | Best supportive care | 320 |
| 5 | Best supportive care | RFA | 82 | Best supportive care | Lenvatinib | 187 |
| 6 | Conventional TACE | Best supportive care | 67 | Conventional TACE | Lenvatinib | 104 |
| 7 | TACE with beads | Surgical resection | 46 | Surgical resection | Sorafenib | 92 |
| 8 | Best supportive care | Liver transplantation | 45 | Sorafenib | Lenvatinib | 87 |
| 9 | Surgical resection | RFA | 37 | Radiation therapy | Best supportive care | 86 |
| 10 | Surgical resection | Best supportive care | 33 | Surgical resection | Best supportive care | 79 |

**Gemini 2.0**

| Rank | BCLC stage A | | | BCLC stage C | | |
|---|---|---|---|---|---|---|
| | Actual treatment | Gemini recommendation | Count | Actual treatment | Gemini recommendation | Count |
| 1 | Conventional TACE | RFA | 606 | Best supportive care | Sorafenib | 939 |
| 2 | Surgical resection | RFA | 465 | Conventional TACE | Sorafenib | 810 |
| 3 | Conventional TACE | Surgical resection | 419 | Conventional TACE | Best supportive care | 184 |
| 4 | RFA | Surgical resection | 249 | Surgical resection | Sorafenib | 170 |
| 5 | Conventional TACE | Best supportive care | 135 | Radiation therapy | Sorafenib | 119 |
| 6 | Best supportive care | Surgical resection | 81 | Best supportive care | Lenvatinib | 94 |
| 7 | Best supportive care | RFA | 79 | Sorafenib | Best supportive care | 72 |
| 8 | RFA | Best supportive care | 67 | Sorafenib | Lenvatinib | 61 |
| 9 | TACE with beads | Surgical resection | 27 | TACE with beads | Sorafenib | 49 |
| 10 | TACE with beads | RFA | 26 | Conventional TACE | Lenvatinib | 47 |

**Claude 3.5**

| Rank | BCLC stage A | | | BCLC stage C | | |
|---|---|---|---|---|---|---|
| | Actual treatment | Claude recommendation | Count | Actual treatment | Claude recommendation | Count |
| 1 | Conventional TACE | RFA | 576 | Best supportive care | Sorafenib | 988 |
| 2 | Surgical resection | RFA | 530 | Conventional TACE | Sorafenib | 816 |
| 3 | Conventional TACE | Surgical resection | 526 | Best supportive care | Lenvatinib | 214 |
| 4 | RFA | Surgical resection | 460 | Sorafenib | Lenvatinib | 160 |
| 5 | Best supportive care | RFA | 124 | Conventional TACE | Lenvatinib | 152 |
| 6 | Best supportive care | Surgical resection | 66 | Surgical resection | Sorafenib | 149 |
| 7 | TACE with beads | RFA | 38 | Radiation therapy | Sorafenib | 116 |
| 8 | Conventional TACE | Best supportive care | 29 | Conventional TACE | Best supportive care | 71 |
| 9 | TACE with beads | Surgical resection | 26 | Surgical resection | Lenvatinib | 52 |
| 10 | Radiation therapy | RFA | 26 | Radiation therapy | Lenvatinib | 46 |

LLM, large language model; TACE, transarterial chemoembolization; RFA, radiofrequency ablation.

the clinical applicability of LLM-derived treatment recommendations in real-world HCC management. In this context, the observed discrepancies between physician decision-making and LLM outputs underscore both the potential strengths and inherent limitations of current LLM-based approaches in contemporary clinical practice.

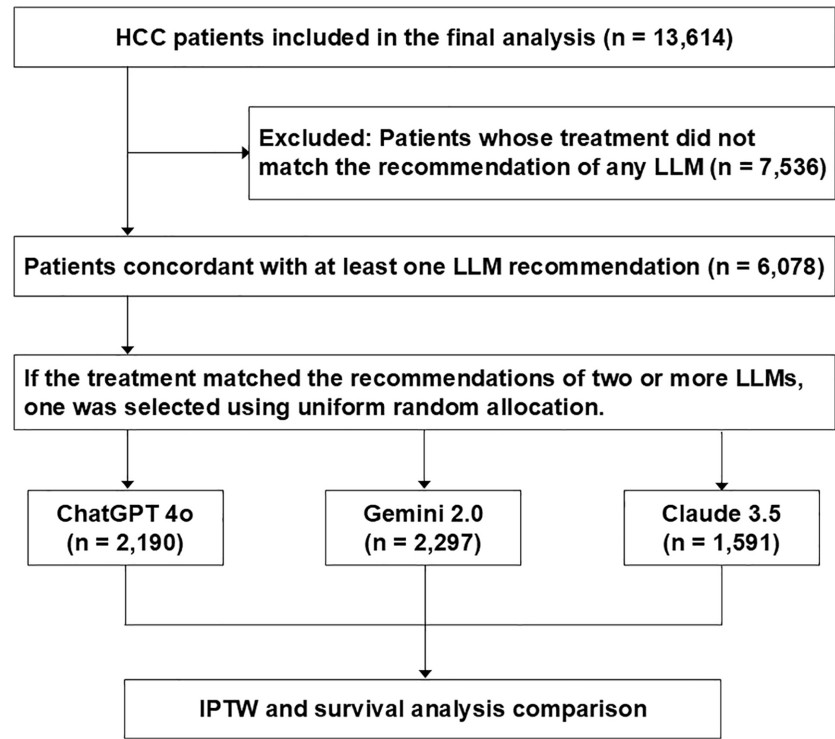

**Fig 5. Flowchart of patient selection and LLM (ChatGPT 4o, Gemini 2.0, and Claude 3.5) allocation strategy.** HCC, hepatocellular carcinoma; LLM, large language model; IPTW, inverse probability of treatment weighting.

Although previous studies suggested that various AI methods can assist the accurate prediction of clinical outcomes in patients with HCC [23–29], the clinical impact of LLM-generated treatment recommendations—particularly in relation to patient survival—has not been established. Overall, the concordance with LLM-generated treatment recommendations was marginally associated with improved OS compared to non-concordant cases. Notably, the survival impact of this concordance differed substantially across BCLC stages. In BCLC-A, patients whose treatments were concordant with LLM recommendations exhibited superior survival outcomes. Conversely, in BCLC-C, patients with non-concordant treatments demonstrated better survival.

We then hypothesized that the variation in survival outcomes reflects differing clinical priorities between LLMs and physicians. Both physicians and LLMs considered the BCLC stage as the most important factor when determining treatment strategies. However, physicians placed greater emphasis on liver function, whereas LLMs prioritized tumor status, and this divergence might contribute to discordant treatment decisions. In BCLC-A, patients in the non-concordant group had poorer liver function, suggesting that clinicians were less likely to pursue curative treatments recommended by LLMs when hepatic reserve was limited. Conversely, in BCLC-C, non-concordant cases tended to have better preserved liver function, indicating a greater willingness among physicians to offer locoregional therapies in selected patients with adequate hepatic function. This may reflect the fact that LLMs base their recommendations strictly on clinical guidelines. Overall, these findings highlight how differing prioritization of clinical factors by LLMs and physicians can lead to discordant treatment decisions and stage-specific survival outcomes, particularly in cases where liver function and tumor status are not aligned. In advanced-stage disease, this stage-dependent pattern was further evident, as concordant cases may have reflected strict adherence to guideline-directed systemic therapy among patients with poor hepatic reserve, leading

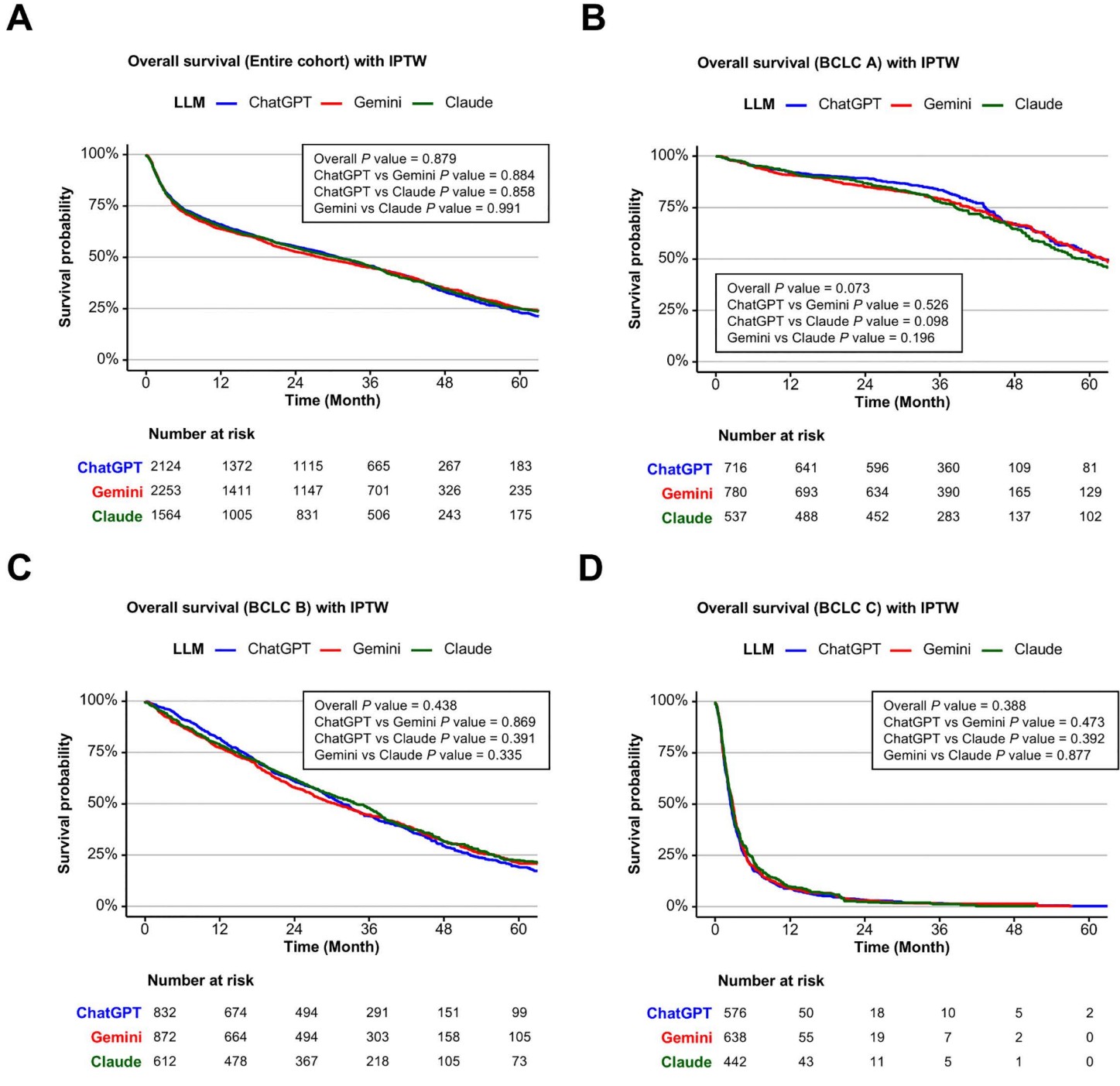

**Fig 6. Kaplan–Meier curves of overall survival according to the large language models (ChatGPT 4o, Gemini 2.0, and Claude 3.5) after IPTW.** **(A)** Entire cohort. **(B)** BCLC stage A. **(C)** BCLC stage B. **(D)** BCLC stage C. P values for overall and pairwise comparisons are shown in each panel.

to selection bias and poorer survival outcomes. This interpretation suggests that the observed associations are driven by clinical heterogeneity rather than by the intrinsic performance of the LLMs. In this context, although metrics such as mean absolute error are useful in prognostic modeling, they are not applicable here because neither the dataset nor the LLM outputs included numerical survival predictions. Our analyses therefore focus on observed outcomes within real-world treatment pathways rather than on quantitative prediction errors.

The differing impact of LLM-physician concordance across BCLC stages reflects the varying complexity of HCC management. Given the intrinsic heterogeneity of the BCLC staging system, several studies have reported discrepancies between real-world treatment practices and guideline recommendations, with measurable effects on survival. In BCLC-A, where treatment guidelines emphasize potentially curative therapies such as resection or ablation, a previous study suggested that guideline-concordant decisions were associated with better survival outcomes [30]. Our finding that LLM-aligned treatments improved survival in this stage supports the idea that consistency with evidence-based recommendations might be advantageous when clinical decision-making is relatively straightforward. Conversely, BCLC-C patients showed better outcomes when their treatment deviated from LLM recommendations. This likely reflects the clinical heterogeneity of advanced HCC, where real-world physicians frequently apply individualized approaches—such as locoregional therapies—in patients with favorable hepatic reserve and limited metastasis, even when guidelines recommend systemic therapy [31,32]. This highlights the limitations of LLMs in accounting for the individualized clinical status that often drive treatment decisions in advanced HCC. These findings demonstrate associations rather than evidence of clinical efficacy, and the remarks on contextual awareness in advanced-stage disease are exploratory and intended for conceptual understanding.

Alternative explanations for the observed discrepancies between physician decisions and LLM-generated treatment recommendations should also be considered. Beyond differences in clinical prioritization, real-world treatment choices are influenced by multiple factors that extend beyond the structured clinical variables used in this study. Patient-related aspects such as individual preferences, surgical eligibility, comorbid conditions, and tolerance for potential complications can substantially shape therapeutic decisions. Moreover, registry-based data cannot account for psychosocial or logistical circumstances that frequently guide multidisciplinary discussions. Therefore, current LLMs depend primarily on textual guideline representations and probabilistic reasoning, which limits their ability to replicate the nuanced, patient-centered judgment that underlies real-world clinical decision-making.

From a clinical perspective, the present findings highlight the potential utility of LLMs in supporting evidence-based treatment decisions, particularly in early-stage HCC where guideline-concordant strategies are clearly defined. The comparable performance observed across ChatGPT 4o, Gemini 2.0, and Claude 3.5 suggests that current LLMs, when provided with structured clinical input, demonstrate a similar capacity to retrieve and apply standardized therapeutic frameworks. This indicates a possible role for LLMs as adjunctive tools in multidisciplinary discussions or in settings with limited access to subspecialty expertise. For instance, LLMs may assist less-experienced clinicians in verifying the concordance of proposed treatment plans with established international guidelines, thereby promoting consistency in clinical practice. However, the applicability of LLMs appears to diminish in more complex scenarios requiring individualized interpretation. Clinical decisions are frequently influenced by nuanced factors—such as patient frailty, portal hypertension, treatment feasibility, and institutional resources—that extend beyond the scope of guideline-based logic. Current LLMs, which rely primarily on generalized textual corpora, might lack the capacity to adequately integrate such context-specific considerations into their recommendations [28].

Despite the strengths of our study, there are several limitations. First, its retrospective observational design and lack of randomization might introduce potential for selection bias. Second, our analysis was restricted to a single national registry spanning over a decade; thus, findings may not fully generalize to other healthcare systems or current treatment paradigm. Second, as this study was based on a nationwide registry, detailed imaging information such as tumor location or resection feasibility was unavailable. This constraint may limit the models' ability to account for spatial factors influencing

treatment eligibility (e.g., resection or ablation). Future multimodal frameworks that integrate imaging and text-based clinical data could further enhance the contextual precision and individualization of LLM-driven treatment recommendations. Third, as the cohort covered 2008–2020, the models reflected guideline recommendations of that era. Emerging therapies—including immunotherapy, combination regimens, and transarterial radioembolization—were not captured, warranting future validation with contemporary data. In addition, the predominance of hepatitis B-related and Child-Pugh A patients reflect the epidemiologic characteristics of HCC in East Asia and may limit direct generalizability to Western populations. Finally, because the study was observational and based on modeled rather than interventional outcomes, causal inference regarding the clinical utility of LLM recommendations cannot be established, underscoring the need for prospective, AI-assisted decision trials compared with multidisciplinary standard care.

In conclusion, this large-scale, real-world analysis is the first to evaluate the clinical relevance of LLM-generated treatment recommendations in HCC. We observed that concordance between LLM suggestions and real-world physician decisions was associated with improved survival in early-stage HCC, whereas such concordance was not beneficial in advanced-stage HCC. These findings suggest that LLMs may reflect established treatment guidelines in straightforward scenarios but lack sufficient contextual awareness for complex clinical decision-making. Accordingly, LLM-generated recommendations should be interpreted with caution and always in conjunction with clinical judgment. Future efforts to enhance model performance may improve the applicability of LLMs as adjunctive tools in HCC care, under appropriate expert supervision.

## Supporting information

**S1 Fig. Overall survival according to concordance between Gemini-recommended and physician-administered treatments in HCC patients.**
(DOCX)

**S2 Fig. Overall survival according to concordance between Claude-recommended and physician-administered treatments in HCC patients.**
(DOCX)

**S3 Fig. One-year landmark overall survival according to concordance between ChatGPT-recommended and physician-administered treatments in patients with HCC.**
(DOCX)

**S4 Fig. One-year landmark overall survival according to concordance between Gemini-recommended and physician-administered treatments in patients with HCC.**
(DOCX)

**S5 Fig. One-year landmark overall survival according to concordance between Claude-recommended and physician-administered treatments in patients with HCC.**
(DOCX)

**S6 Fig. Covariate balance before and after weighting by LLM and BCLC stage.**
(DOCX)

**S7 Fig. Simplified decision trees for treatment recommendations by clinicians and LLMs.**
(DOCX)

**S8 Fig. Concordance between ChatGPT-recommended treatments and actual clinical decisions in HCC patients.**
(DOCX)

**S9 Fig. Concordance between Gemini-recommended treatments and actual clinical decisions in HCC patients.**
(DOCX)

**S10 Fig. Concordance between Claude-recommended treatments and actual clinical decisions in HCC patients.**
(DOCX)

**S11 Fig. Subgroup concordance analysis between Gemini-based treatment suggestions and clinical practice across BCLC stages.**
(DOCX)

**S12 Fig. Subgroup concordance analysis between Claude-based treatment suggestions and clinical practice across BCLC stages.**
(DOCX)

**S13 Fig. Subgroup concordance analysis between LLM-based treatment suggestions and clinical practice across BCLC stages.**
(DOCX)

**S14 Fig. Kaplan–Meier curves of overall survival according to the large language models before IPTW.**
(DOCX)

**S1 Table. CHART checklist.**
(DOCX)

**S2 Table. Baseline clinical characteristics according to concordance between physician decisions and Gemini-generated treatment recommendations.**
(DOCX)

**S3 Table. Baseline clinical characteristics according to concordance between physician decisions and Claude-generated treatment recommendations.**
(DOCX)

**S4 Table. Univariate analyses of overall survival in HCC patients according to BCLC stage.**
(DOCX)

**S5 Table. Era-stratified sensitivity analysis of survival outcomes by BCLC stage and LLM model.**
(DOCX)

**S6 Table. Inverse probability weighted cox proportional hazards model for overall survival according to LLM recommendation adherence (BCLC Stage A and C).**
(DOCX)

**S7 Table. Doubly robust (Augmented IPTW) cox proportional hazards model for overall survival according to LLM recommendation adherence.**
(DOCX)

**S8 Table. Baseline clinical characteristics according to concordance between physician decisions and ChatGPT-generated treatment recommendations in BCLC stage A.**
(DOCX)

**S9 Table. Baseline clinical characteristics according to concordance between physician decisions and ChatGPT-generated treatment recommendations in BCLC stage C.**
(DOCX)

**S10 Table. Baseline clinical characteristics according to concordance between physician decisions and Gemini-generated treatment recommendations in BCLC stage A.**
(DOCX)

**S11 Table. Baseline clinical characteristics according to concordance between physician decisions and Gemini-generated treatment recommendations in BCLC stage C.**
(DOCX)

**S12 Table. Baseline clinical characteristics according to concordance between physician decisions and Claude-generated treatment recommendations in BCLC stage A.**
(DOCX)

**S13 Table. Baseline clinical characteristics according to concordance between physician decisions and Claude-generated treatment recommendations in BCLC stage C.**
(DOCX)

**S14 Table. Top 10 most frequent disagreement patterns between LLM recommendations and actual treatments in the entire cohort.**
(DOCX)

**S15 Table. Median overall survival according to concordance between LLM recommendations and actual treatment across BCLC stages.**
(DOCX)

**S16 Table. Stage-specific adherence and treatment-tier transition patterns across large language models.**
(DOCX)

**S17 Table. Within-stage IPTW-adjusted effects of tier escalation on overall survival across large language models.**
(DOCX)

**S18 Table. Baseline clinical characteristics of HCC patients according to the LLM model before IPTW.**
(DOCX)

**S19 Table. Baseline clinical characteristics of HCC patients according to the LLM model after IPTW.**
(DOCX)

**S1 Data. Raw outputs of LLM-generated treatment recommendations.**
(XLSX)

## Acknowledgments

The clinical data of the present study were provided by the Korean Liver Cancer Study Group and Korean Central Cancer Registry of the Ministry of Health and Welfare, Republic of Korea.

## Author contributions

**Conceptualization:** Keungmo Yang, Ji Won Han.

**Data curation:** Keungmo Yang, Jaejun Lee, Ji Won Han.

**Formal analysis:** Keungmo Yang, Jaejun Lee, Ji Won Han.

**Funding acquisition:** Ji Won Han.

**Investigation:** Keungmo Yang, Jeong Won Jang, Pil Soo Sung, Ji Won Han.

**Methodology:** Keungmo Yang, Pil Soo Sung, Ji Won Han.

**Project administration:** Keungmo Yang, Ji Won Han.

**Resources:** Keungmo Yang.

**Software:** Keungmo Yang, Jaejun Lee.

**Supervision:** Jeong Won Jang, Ji Won Han.

**Validation:** Keungmo Yang, Jaejun Lee, Pil Soo Sung, Ji Won Han.

**Visualization:** Keungmo Yang.

**Writing – original draft:** Keungmo Yang, Ji Won Han.

**Writing – review & editing:** Keungmo Yang, Jaejun Lee, Jeong Won Jang, Pil Soo Sung, Ji Won Han.

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
