## [Editor Report · Decision Letter 0]

10 Oct 2025

Dear Dr Han,

Thank you for submitting your manuscript entitled "Evaluating the Clinical Utility of Large Language Models for Hepatocellular Carcinoma Treatment Recommendations" for consideration by PLOS Medicine.

Your manuscript has now been evaluated by the PLOS Medicine editorial staff as well as by an academic editor with relevant expertise and I am writing to let you know that we would like to send your submission out for external peer review.

For clinical studies, please upload a copy of your trial study protocol as a supporting information file. The study protocol should be the version submitted for approval to the institutional review board or ethics committee, should include any amendments to the study protocol, as well as the date of their approval by the institutional review or ethics committee. Please also detail any deviations from the study protocol in the Methods section of your manuscript. The editors will consider the protocol and study conduct prior to a final decision for external review.

Please re-submit your manuscript within two working days, i.e. by Oct 14 2025 11:59PM.

Kind regards,

Heather Van Epps, PhD

Senior Editor

PLOS Medicine

---

## [Decision Letter · Decision Letter 1]

20 Oct 2025

Dear Dr Han,

Many thanks for submitting your manuscript "Evaluating the Clinical Utility of Large Language Models for Hepatocellular Carcinoma Treatment Recommendations" (PMEDICINE-D-25-03475R1) to PLOS Medicine. The paper has been reviewed by subject experts, a statistician and an academic editor; their comments are included below and can also be accessed here: [LINK]

As you will see, while the reviewers find the work interesting, they raise important concerns and identify the need for additional analyses to consider the study further. In particular the reviewers require additional information pertaining to the prompts used, the method of determining concordance between physician and LLM, consideration of potential bias associated with use of overall survival as outcome, and that you address discordance between physician recommendation and actual treatment. Reviewer 3 requests additional analyses and consideration of whether the study adequately reflects current treatment recommendations.

After discussing the paper with the editorial team and an academic editor with relevant expertise, I'm pleased to invite you to revise the paper in response to the reviewers' comments. We plan to send the revised paper to some or all of the original reviewers, and we cannot provide any guarantees at this stage regarding publication.

Please further note that any claims of clinical utility of LLMs must be removed given that the study is not a prospective trial. The Conclusions section of the Abstract must be revised accordingly, as well as the framing of concordance of physician and LLM recommendations in the Abstract.

The technological limitations of the methods must be stated in the Abstract (Methods & Findings section), and the overall limitations must be included in the last section of the Author Summary.

New code generated for this study must be deposited in a publicly accessible database. Please also refer to our Data Availability requirements and specify all restrictions to data access and the process for data access. Please note that 'upon reasonable request' is not an approved process for data access.

Please include a completed CHART and/or TRIPOD-LLM checklist.

We ask that you submit your revision by Nov 10 2025 11:59PM. However, if this deadline is not feasible, please contact me by email, and we can discuss a suitable alternative.

Don't hesitate to contact me directly with any questions (afarrell@plos.org).

Best regards,

Alison

Alison Farrell, Ph.D.

Senior Editor

PLOS Medicine

afarrell@plos.org

Comments from the academic editor:

1. The inputs for the LLM models are unclear and likely insufficient to make accurate treatment recommendations (which drove some of the concordance). For example, eligibility for ablation and resection can be influenced by tumor location, which would be difficult for the models to take into account without being fed full images.

2. BCLC staging is heterogeneous and several studies have shown discordance between treatment practices and guideline recommendations (including impact on survival).

3. The cohort and guideline recommendations are now older (with both AASLD and EASL being updated in the past 1-2 years), and several treatment regimens now evolving over time including introduction of immunotherapy and increasing use of TARE.

4. Higher proportion of HBV and Child Pugh A patients decreases generalizability to many Western centers.

5. Table 3 is informative but it would be better if there were details of why treatments differed. For example, why were 843 patients treated with TACE instead of resection? Without knowing if this was appropriate (e.g. portal hypertension or tumor location) or inappropriate (lack of seeing a surgeon), it's hard to understand the clinical relevance.

Comments from the reviewers:

Reviewer #1: "Evaluating the Clinical Utility of Large Language Models for Hepatocellular Carcinoma Treatment Recommendations" explores the use of large language models (LLMs), for the management of hepatocellular carcinoma (HCC) treatment. This was done by comparing LLM-generated recommendations against actual clinical decisions and survival outcomes (overall survival; OS), on some 13000 patients from the Korean Primary Liver Cancer Registry. It was concluded that LLM recommendations had clinical utility in early-stage HCC, but not advanced-stage HCC.

While the findings are informative towards the potential use of LLMs as an adjunctive tool towards HCC management, some initial concerns might be addressed:

1. In the Generation and Evaluation of LLM-Based Treatment Recommendations section, it is stated that patients were classified as "matched" if the LLM recommendation corresponded to the actual treatment received. It might be clarified as to whether an exact textual match was required for the recommendation, and if not, how non-exact matches were converted.

2. Related to the above, the standardized prompts used for the LLMs should be included in supplementary material, possibly with example inputs and generated outputs (and corresponding physician recommendations). In particular, did the LLM prompts constrain the possible recommendations?

3. In the Impact of concordance between LLM-physician on OS section, it is stated that in general, patients whose treatments matched LLM recommendations (i.e. the LLM assessment and original clinical assessment are in agreement) exhibited significantly better survival outcomes, particularly for early-stage BCLC-A and BCLC-B patients. However, for advanced-stage BCLC-C patients, matched cases had significantly worse OS (HR=2.27) compared to mismatched cases.

Firstly, the claim of clinical utility for LLM in early-stage cases but not advanced-stage cases should be further justified. In particular, the counterfactual outcome where the LLM recommendation is applied (instead of the original recommendation in real life) remains unknown; considering this, assertions about the true utility of the LLM recommendations in mismatched cases (relating to claims that "...they lack contextual awareness for complex clinical situations requiring individualized care") may not be warranted. This issue should be carefully considered if deemed appropriate.

4. Related to the above, the observation that LLM and physician concordance for advanced-stage patients was associated with worse OS might be analyzed further. The following sections (and Table 3) attempt to explain causes and patterns of discordance between LLMs and physicians, but do not appear to discuss and analyze patterns of concordance, especially for BCLC-C cases. This is important since the presumed use of LLMs as an adjunctive tool would appear to involve identifying cases for deeper assessment by physicians.

5. The use of overall survival (OS) as a metric for quantifying the clinical relevance of LLM recommendations may have caveats. In particular, it does not appear to take into account the actual expected survival outcome of the patient, given their specific circumstances (e.g. liver function, tumor factors, etc.). As such, if an LLM (or any other statistical or machine learning model) biases its recommendations towards patient profiles with higher survival rates, then its OS metric for matched cases would automatically improve as a result.

As a simple concrete example, consider a HCC population that has 50% of cases with surgical resection as best treatment (with a 90% survival rate), and the other 50% of cases with best supportive case as best treatment (with a 10% survival rate). Then, an LLM that simply always recommends surgical resection would achieve a very high OS for matched cases, whether or not the recommendation is logically justified (which is often not true, since this LLM always gives the same recommendation). However, this LLM would be concluded to have high clinical utility from the OS outcome.

If this understanding is correct, authors might consider also incorporating the actual expected survival outcome of patients, in analyzing the clinical utility of the LLMs.

6. Related to the above, complete matrices showing the prevalence and OS of all possible recommendation combinations (similar to Figure 4c) might be included as a figure. The OS for each actual treatment might also be included as a table.

7. It might be clarified as to whether physician treatment recommendations are always followed, or whether other considerations (e.g. cost, patient choice) may result in physician recommendation and actual treatment being discordant. If the latter, the prevalence of such discordant treatments might be commented on.

8. In the Factors underlying LLM-physician discordance section, it is stated that decision tree models were constructed to assess key clinical variables influencing treatment selection in both physician decisions and LLM-generated recommendations. However, Figure 3 appears to show feature importances of various factors. The construction methodology of the decision trees for both LLM and physician recommendations should thus be described in detail, possibly in supplementary material. If possible, example decision trees and their corresponding recommendation should also be provided.

Reviewer #2: The authors task three generative AI chatbots with providing treatment advice for hepatocellular carcinoma patients, comparing this output to actual decisions made by the patients' physicians. This is an interesting application and of interest to clinicians and researchers wondering about the potential applications of AI. Reporting could be strengthened using CHART (see point 1), and more detail is necessary in the Methods section. Other than this, my comments are mostly minor.

1. The CHART tool (https://www.equator-network.org/reporting-guidelines/reporting-guideline-for-chatbot-health-advice-studies-the-chart-statement/) has been designed for studies of generative AI providing advice to clinicians or patients. The manuscript would be strengthened by adhering to this. Guidance is provided in any of six peer reviewed articles, and a website is available to automatically generate a checklist.

2. In the Methods, the models should be explicitly described as proprietary and 'out-of-the-box'/unmodified (presuming the authors made no fine-tuning or other modification beyond the information in the prompts described).

3. Was any formal prompt engineering process undertaken, or how were prompts designed? Who designed them, were multiple prompts tested, and how was a final prompt selected? The AASLD and EASL guidelines should also be cited.

4. More information regarding evaluation is necessary in the Methods, perhaps in an additional subsection. Who decided whether treatment was concordant with actual treatment; did more than one judge conduct assessment; and how were any discrepencies resolved? Did treatment have to match exactly, or was the same agent / type of regime / duration of treatment sufficient? In short, precisely what counted as matching or mismatching, and why?

5. The process of modelling overall survival following LLM recommendations is not clear--and currently reads almost as though patients underwent treatment based on the LLM suggestion (which is not the case). It must be crystal clear that survival analyses stratified patients based on concordance between LLM and physician decisions.

6. In the Results, references to 'curative therapy' should read 'potentially curative therapy'.

7. In the Results subsection "Factors underlying LLM-physician discordance' there is too much speculation, generally at the end of each paragraph. These speculations as to why discrepancies are observed should be explained in the Discussion as there are not proven or disproven by the study.

8. The Discussion does not explore alternative hypotheses for why discrepancies between physicians and generative AI is observed. For instance, patients may choose not to or to have surgery if offered, which would be expected to lead to discrepancy with guideline recommendations.

9. The limitations paragraph is far too brief. The observational design precludes any causal inference or conclusions regarding the utility of generative AI due to selection bias as well as apple to oranges comparisons (actual versus modelled survival) and neglect of patient opinions/decisions regarding treatment. A call for prospective studies of AI-enhanced decision-making versus conventional care directed by the multidisciplinary team would be appropriate.

Reviewer #3: This large registry study (2008-2020) compares LLM‑generated, guideline‑based treatment suggestions for HCC with real‑world management and outcomes. While stage‑stratified results are intriguing, the current design cannot disentangle adherence effects from baseline differences and secular changes in practice.

Major comments

1) Confounding by indication & time‑related bias (primary concern).

The survival gap between "match" and "mismatch" groups likely reflects baseline imbalances and feasibility rather than the act of following an LLM recommendation. Please (a) define a clear time‑zero and a prespecified treatment window (e.g., 60-90 days) and use either a landmark analysis (follow‑up starts at the landmark) or a time‑dependent treatment model to avoid immortal‑time bias; (b) reframe analyses within each BCLC stage and recommended therapy as the effect of adhering to that therapy; (c) estimate stabilized IPTW (or overlap weights) for adherence using only pre‑treatment covariates (age, ECOG, Child‑Pugh/ALBI, portal‑hypertension surrogates, tumor burden, AFP, calendar year, and—if available—center). Show balance diagnostics (SMD <0.10) and fit a weighted Cox (Aalen-Johansen if competing risks are relevant). Provide sensitivity analyses with doubly‑robust estimators (AIPW/TMLE). Temper causal language accordingly.

2) Clinical perspective: guidelines vs individualised care, and therapeutic hierarchy.

Non‑adherence to AASLD/BCLC is not inherently detrimental; in complex, multiparametric HCC, individualised decisions outside rigid algorithms can yield better outcomes (e.g., Lancet Oncol 2023; doi: 10.1016/S1470-2045(23)00186-9). Your own findings suggest that the apparent advantage of certain LLM cohorts stems less from "following guidelines" and more from recommending higher‑tier, curative‑intent options (resection, transplant, ablation) relative to real‑world choices.

Actionable analyses:

Classify treatments into ordered tiers (curative‑intent: resection/transplant/ablation; non‑curative: TACE; systemic; best supportive care).

Decompose effects into (i) adherence to the specific recommendation and (ii) tier escalation (LLM‑recommended therapy higher vs equal/lower tier than real‑world).

Within‑stage, estimate the effect of tier escalation with IPTW and report both survival and treatment‑receipt outcomes (e.g., odds of receiving curative‑intent therapy). This will clarify whether LLMs are useful because they privilege higher‑value therapies, not simply because they mirror guidelines.

3) Secular trends and center effects.

Given major shifts in locoregional techniques and systemic therapy across 2008-2020, adjust for calendar time (splines or eras) and, if available, center; add era‑stratified sensitivity analyses, especially in BCLC‑C.

4) Minimal transparency/reproducibility fixes.

Align the Data & Code Availability statement with PLOS policy; deposit de‑identified analysis‑ready data (or an auditable access pathway) and all scripts/prompts/parameters.

Document LLM versioning (model versions, inference dates, decoding parameters, number of runs/seeds) and provide raw outputs to ensure reproducibility.

Minor

1) Reconcile any table-figure HR inconsistencies in BCLC‑B; improve figure resolution and provide vector graphics with numbers‑at‑risk.

2) Standardise TACE terminology across text and figures.

---

* Please upload any figures associated with your paper as individual TIF or EPS files with 300dpi resolution at resubmission; please read our figure guidelines for more information on our requirements: http://journals.plos.org/plosmedicine/s/figures. While revising your submission, we strongly recommend that you use PLOS's NAAS tool (https://ngplosjournals.pagemajik.ai/artanalysis) to test your figure files. NAAS can convert your figure files to the TIFF file type and meet basic requirements (such as print size, resolution), or provide you with a report on issues that do not meet our requirements and that NAAS cannot fix.

After uploading your figures to PLOS's NAAS tool - https://ngplosjournals.pagemajik.ai/artanalysis, NAAS will process the files provided and display the results in the "Uploaded Files" section of the page as the processing is complete.

If the uploaded figures meet our requirements (or NAAS is able to fix the files to meet our requirements), the figure will be marked as "fixed" above. If NAAS is unable to fix the files, a red "failed" label will appear above.

When NAAS has confirmed that the figure files meet our requirements, please download the file via the download option, and include these NAAS processed figure files when submitting your revised manuscript.

* [EDITOR: CHECK FINANCIAL DISCLOSURES, COI, DAS, AND ETHICS STATEMENTS AND INCLUDE ANY NECESSARY REQUESTS]

* Please ensure that the study is reported according to the [XXXX] guideline and include the completed [XXXX] checklist as Supporting Information. When completing the checklist, please use section and paragraph numbers, rather than page numbers. Please add the following statement, or similar, to the Methods: "This study is reported as per [XXXX] guideline (S1 Checklist)."

FIGURES AND TABLES

SUPPLEMENTARY MATERIAL

REFERENCES

[STUDY TYPE-SPECIFIC REQUESTS - DELETE SECTIONS AS NECESSARY]

RCTs [REFER TO RCT CHECKLIST AND MEETING NOTES FOR DETAILS TO ADD]

* PLOS Medicine requires that all trials be prospectively registered in one of registries recognized by WHO. Please ensure that study registration details are included in the Methods section.

* Please structure the Methods section using the following sub-headings: Study design and participants, Randomization and masking, Procedures, Outcomes, Statistical analysis.

* The following outcomes measures [ADD DETAILS AS NEEDED OR DELETE BULLET POINT] appear to differ between the submitted manuscript and the protocol [and/or trial registry]. Please clarify and explain all discrepancies between the paper and protocol. If the outcomes were not prespecified in the protocol, please define them in the Methods (Outcomes section) as post hoc and explain why they were added. Post-hoc comparisons should be presented as hypothesis generating rather than conclusive.

* Please ensure that all prespecified outcomes (primary, secondary, and exploratory) are listed in the Methods/Outcomes section and indicate whether there are outcomes that are not presented in the current report.

* Please specify the dates (Month Day, Year) during which study enrollment and follow up occurred.

* Please include absolute numbers wherever you report percentages; eg, n/N (%)

* Please present the safety data for the study including numbers of specific events and whether or not adverse events are thought to be related to treatment. AEs should be reported in the abstract, per CONSORT and CONSORT-Harms.

* Please complete the CONSORT checklist (https://www.equator-network.org/reporting-guidelines/consort/) and ensure that all components of CONSORT are present in the manuscript, including how randomization was performed, allocation concealment, blinding of intervention, definition of lost to follow-up, power statement. When completing the checklist, please use section and paragraph numbers, rather than page numbers.

* Please report your abstract according to CONSORT for abstracts, following the PLOS Medicine abstract structure (Background, Methods and Findings, Conclusions) https://www.equator-network.org/reporting-guidelines/consort-abstracts/

* If your trial had to undergo important modifications in response to extenuating circumstances, please complete the CONSERVE-CONSORT checklist and provide in your Supporting Information; (https://www.equator-network.org/reporting-guidelines/guidelines-for-reporting-trial-protocols-and-completed-trials-modified-due-to-the-covid-19-pandemic-and-other-extenuating-circumstances-the-conserve-2021-statement/). When completing the checklist, please use section and paragraph numbers, rather than page numbers.

* In keeping with our commitment to Open Science, please include the study protocol document and analysis plan (including any amendments) as Supporting Information to be published with the manuscript if accepted.

* Please note that PLOS Medicine requires prospective, public registration of a data sharing plan (as part of mandatory clinical trials registration) for all clinical trials that began enrollment on or after January 1, 2019, in accordance with ICMJE requirements.

OBSERVATIONAL STUDIES

* Abstract: Please include the study design, population and setting, number of participants, years during which the study took place (enrollment and follow up), length of follow up, and main outcome measures.

* Please ensure that the study is reported according to the STROBE (or appropriate STOBE extension) guideline (available from: https://www.equator-network.org/reporting-guidelines/strobe) and include the completed STROBE (or STROBE extension) checklist as Supporting Information. Please add the following statement, or similar, to the Methods: "This study is reported as per the Strengthening the Reporting of Observational Studies in Epidemiology (STROBE) guideline (S1 Checklist)." When completing the checklist, please use section and paragraph numbers, rather than page numbers.

* [FOR POPULATION HEALTH/REGISTRY STUDIES] Please ensure that the study is reported according to the RECORD guideline (available from https://www.record-statement.org) and include the completed checklist as Supporting Information. Please add the following statement, or similar, to the Methods: "This study is reported as per the Reporting of Studies Conducted using Observational Routinely-Collected Data (RECORD) guideline (S1 Checklist)." When completing the checklist, please use section and paragraph numbers, rather than page numbers.

* [FOR POPULATION HEALTH ESTIMATES] Please ensure that the study is reported according to the GATHER statement (available from https://www.equator-network.org/reporting-guidelines/gather-statement) and include the completed checklist as Supporting Information. Please add the following statement, or similar, to the Methods: "This study is reported as per the Guidelines for Accurate and Transparent Health Estimates Reporting (GATHER) statement (S1 Checklist)." When completing the checklist, please use section and paragraph numbers, rather than page numbers.

* [FOR MEDIATION ANALYSES] We recommend that the study is reported according to the AGReMA statement (https://agrema-statement.org/#:~:text=AGReMA%20is%20an%20evidence%2D%20and,randomised%20trials%20and%20observational%20studies) and include the completed checklist as Supporting Information. Please add the following statement, or similar, to the Methods: "This study is reported as per the Guideline for Reporting Mediation Analyses (AGReMA) statement (S1 Checklist)." When completing the checklist, please use section and paragraph numbers, rather than page numbers.

* For all observational studies, in the manuscript text, please indicate: (1) the specific hypotheses you intended to test, (2) the analytical methods by which you planned to test them, (3) the analyses you actually performed, and (4) when reported analyses differ from those that were planned, transparent explanations for differences that affect the reliability of the study's results. If a reported analysis was performed based on an interesting but unanticipated pattern in the data, please be clear that the analysis was data driven.

* Please state in the Methods section whether the study had a prospective protocol or analysis plan. If a prospective analysis plan (from your funding proposal, IRB or other ethics committee submission, study protocol, or other planning document written before analyzing the data) was used in designing the study, please include the relevant document(s) with your revised manuscript as a Supporting Information file to be published alongside your study and cite it in the Methods section. A legend for this file should be included at the end of your manuscript. If no such document exists, please make sure that the Methods section transparently describes when analyses were planned, and when/why any data-driven changes to analyses took place. Changes in the analysis, including those made in response to peer review comments, should be identified as such in the Methods section of the paper, with rationale.

MODELLING STUDIES

The following list is derived from Geoffrey P Garnett, Simon Cousens, Timothy B Hallett, Richard Steketee, Neff Walker. Mathematical models in the evaluation of health programmes. (2011) Lancet DOI:10.1016/S0140-6736(10)61505-X:

* If pertinent, please provide a diagram that shows the model structure, including how the natural history of the disease is represented, the process and determinants of disease acquisition, and how the putative intervention could affect the system.

* Please provide a complete list of model parameters, including clear and precise descriptions of the meaning of each parameter, together with the values or ranges for each, with justification or the primary source cited and important caveats about the use of these values noted.

* Please provide a clear statement about how the model was fitted to the data, including goodness-of-fit measure, the numerical algorithm used, which parameter varied, constraints imposed on parameter values, and starting conditions.

* For uncertainty analyses, please state the sources of uncertainties quantified and not quantified [can include parameter, data, and model structure].

* Please provide sensitivity analyses to identify which parameter values are most important in the model. Uncertainty estimates seek to derive a range of credible results on the basis of an exploration of the range of reasonable parameter values. The choice of method should be presented and justified.

* Please discuss the scientific rationale for the choice of model structure and identify points where this choice could influence conclusions drawn. Please also describe the strength of the scientific basis underlying the key model assumptions.

* For studies that develop a prediction model or evaluate its performance, please ensure that the study is reported according to the TRIPOD statement (https://www.equator-network.org/reporting-guidelines/tripod-statement) and include the completed checklist as Supporting Information. Please add the following statement, or similar, to the Methods: "This study is reported as per the Transparent Reporting of a Multivariable Prediction Model for Individual Prognosis Or Diagnosis (TRIPOD) statement (S1 Checklist)." For studies using machine learning, please use the TRIPOD-AI checklist. When completing the checklist, please use section and paragraph numbers, rather than page numbers.

DIAGNOSTIC STUDIES

* Please ensure that the study is reported according to the STARD guideline (https://www.equator-network.org/reporting-guidelines/stard/) and include the completed STARD checklist as Supporting Information. Please add the following statement, or similar, to the Methods: "This study is reported as per the Standards for Reporting of Diagnostic Accuracy (STARD) guideline (S1 Checklist)." When completing the checklist, please use section and paragraph numbers, rather than page numbers.

* Please structure your Abstract according to STARD for Abstracts (https://www.equator-network.org/reporting-guidelines/stard-abstracts/).

* Please structure the Methods section using the following sub-headings: Study design, Participants, Test methods, Analysis.

* Please include a diagram to describe the flow of participants through the study (typically figure 1).

MENDELIAN RANDOMIZATION STUDIES

* Please ensure that the study is reported according to the STROBE-MR guideline (https://www.equator-network.org/reporting-guidelines/strobe/) and include the completed STROBE-MR checklist as Supporting Information. Please add the following statement, or similar, to the Methods: "This study is reported as per the Strengthening the Reporting of Observational Studies in Epidemiology (STROBE) guideline, specific for mendelian randomization (S1 Checklist)." When completing the checklist, please use section and paragraph numbers, rather than page numbers.

* In the Introduction, please describe the exposure and the evidence for a potential causal relationship between exposure and outcome.

* In the Methods, please explicitly state the 3 core instrumental variable assumptions for the main analysis (relevance, independence, and exclusion restriction), as well assumptions for any additional or sensitivity analysis.

* In the Methods, please describe the MR estimator (e.g., 2-stage least squares, Wald ratio) and related statistics. Detail the included covariates and, in case of 2-sample MR, whether the same covariate set was used for adjustment in the 2 samples.

* If you are presenting an instrumental variable estimate, please compare this to the conventional observational estimate.

* Report the associations between genetic variant and exposure and between genetic variant and outcome, preferably on an interpretable scale.

* Report MR estimates of the relationship between exposure and outcome and the measures of uncertainty from the MR analysis, on an interpretable scale, such as odds ratio or relative risk per SD difference.

* If relevant, please consider translating estimates of relative risk into absolute risk for a meaningful time period.

* Please consider including plots to visualize results (e.g., forest plot, scatterplot of associations between genetic variants and outcome vs between genetic variants and exposure).

SURVEY-BASED STUDIES

* Please ensure that the study is reported according to the CROSS guideline (https://www.equator-network.org/reporting-guidelines/a-consensus-based-checklist-for-reporting-of-survey-studies-cross/) and include the completed CROSS checklist as Supporting Information. Please add the following statement, or similar, to the Methods: "This study is reported as per A Consensus-Based Checklist for Reporting of Survey Studies (CROSS) guideline (S1 Checklist)." When completing the checklist, please use section and paragraph numbers, rather than page numbers.

* Please report your survey response rates according to AAPOR recommendations (https://aapor.org/standards-and-ethics/best-practices/)

* Please define how the population surveyed was sampled.

* Please compare characteristics of respondents and nonrespondents if possible.

* If sequential waves of the survey were sent, please specify whether the characteristics of respondents changed over time or remained constant.

* Please include the survey response rate in the Abstract.

* Please include a copy of the survey in the supplementary files.

SYSTEMATIC REVIEWS & META-ANALYSES

* Please report your SR/MA according to the PRISMA guidelines provided at the EQUATOR site. http://www.equator-network.org/reporting-guidelines/prisma/. Please provide the completed PRISMA checklist as Supporting Information. When completing the checklist, please use section and paragraph numbers, rather than page numbers. Please add the following statement, or similar, to the Methods: "This study is reported as per the Preferred Reporting Items for Systematic Reviews and Meta-Analyses (PRISMA) guideline (S1 Checklist)."

* Abstract: Please report your abstract according to PRISMA for abstracts (https://doi.org/10.1371/journal.pmed.1001419) following the PLOS Medicine abstract structure (Background, Methods and Findings, Conclusions). Please ensure you provide dates of search, data sources, number of studies included, types of study designs included, eligibility criteria, and synthesis/appraisal methods.

* Please note that we expect searches to be updated to within 6 months of the time of submission.

QUALITATIVE STUDIES

* Please report your qualitative study according to the appropriate study design provided at (http://www.equator-network.org/?post_type=eq_guidelines&eq_guidelines_study_design=qualitative-research&eq_guidelines_clinical_specialty=0&eq_guidelines_report_section=0&s=) and provide the relevant completed checklist as a supplemental file. In the checklist, please include sufficient text excerpted from the manuscript to explain how you accomplished all applicable items. When completing checklists, please use section and paragraph numbers, rather than page numbers.

* We recommend that authors use the COREQ checklist, or other relevant checklists listed by the Equator Network, such as the SRQR, to ensure complete reporting (see: http://www.equator-network.org/?post_type=eq_guidelines&eq_guidelines_study_design=qualitative-research&eq_guidelines_clinical_specialty=0&eq_guidelines_report_section=0&s=). Please add the following statement, or similar, to the Methods: "This study is reported as per the Consolidated criteria for reporting qualitative research (COREQ): a 32-item checklist for interviews and focus groups (S1 Checklist)."

* In general, we expect qualitative studies to include the following: 1) defined objectives or research questions; 2) description of the sampling strategy, including rationale for the recruitment method, participant inclusion/exclusion criteria and the number of participants recruited; 3) detailed reporting of the data collection procedures; 4) data analysis procedures described in sufficient detail to enable replication; 5) a discussion of potential sources of bias; and 6) a discussion of limitations.

HEALTH ECONOMICS / COST-EFFECTIVENESS STUDIES

* Please ensure that the study is reported according to the CHEERS guideline (available from: https://www.equator-network.org/reporting-guidelines/cheers) and include the completed checklist as Supporting Information. Please add the following statement, or similar, to the Methods: "This study is reported as per the Strengthening the Consolidated Health Economic Evaluation Reporting Standards 2022 (CHEERS 2022) Statement (S1 Checklist)." When completing the checklist, please use section and paragraph numbers, rather than page numbers.

---

## [Decision Letter · Decision Letter 2]

19 Nov 2025

Dear Dr. Han,

Thank you very much for re-submitting your manuscript "Evaluating the Clinical Utility of Large Language Models for Hepatocellular Carcinoma Treatment Recommendations" (PMEDICINE-D-25-03475R2) for review by PLOS Medicine.

I have discussed the paper with my colleagues and the academic editor and it was also seen again by 3 reviewers. I am pleased to say that provided the remaining editorial and production issues are dealt with we are planning to accept the paper for publication in the journal.

We ask you to address the remaining concerns of reviewers 1 and 2 in a revised manuscript, and please include a point-by-point response to the reviewers' comments and the editorial requests at the end of the email.

[LINK]

We look forward to receiving the revised manuscript by Nov 25 2025 11:59PM.   

Sincerely,

Alison Farrell, Ph.D.

Senior Editor 

PLOS Medicine

plosmedicine.org

Requests from Editors:

General:

The Author Summary repeats phrasing in the Abstract. Please revise the Author summary and make accessible to the general reader.

* Please confirm that your title complies with PLOS Medicine's style. Your title must be nondeclarative and not a question. It should begin with main concept if possible. "Effect of" should be used only if causality can be inferred, i.e., for an RCT. Please place the study design ("A randomized controlled trial," "A retrospective study," "A modelling study," etc.) in the subtitle (ie, after a colon).

* Please confirm that your abstract complies with our requirements, including format (three sections: Background, Methods and Findings, and Conclusions) and providing all the information relevant to this study type https://journals.plos.org/plosmedicine/s/submission-guidelines#loc-abstract

* Please ensure that the Introduction ends with a clear description of the study question or hypothesis.

* Please ensure that all abbreviations are defined at first use throughout the text.

* Please confirm that all numbers presented in the abstract are present and identical to numbers presented in the main manuscript text.

* Please include the URLs for the funders in the funding statement.

* Statistical reporting:

- Please report statistical information as follows to improve clarity for the reader ""22% (95% CI [13,28]; p</=)"".

- Please separate upper and lower bounds with commas instead of hyphens as the latter can be confused with reporting of negative values.

- Please repeat statistical definitions (HR, CI etc.) for each set of parentheses."

* In the abstract, please include the important dependent variables that are adjusted for in the analyses.

"* PLOS defines the “minimal data set” to consist of the data set used to reach the conclusions drawn in the manuscript with related metadata and methods, and any additional data required to replicate the reported study findings in their entirety. Authors do not need to submit their entire data set, or the raw data collected during an investigation. Please submit the following data:

The values behind the means, standard deviations and other measures reported;

The values used to build graphs;

The points extracted from images for analysis.

* The Data Availability Statement (DAS) in the manuscript metadata requires revision as it does not match the restricted data availability stated in the manuscript. If the data are not freely available, please describe briefly the ethical, legal, or contractual restriction that prevents you from sharing it. Please also include an appropriate contact (email address) for inquiries (this cannot be a study author).

Figures and Tables:

* Please define all elements of box plots in the figure caption - center line, box limits and whiskers.

* Please provide titles and legends for all figures and tables (including those in Supporting Information files). Please define all acronyms used in each figure or table in its corresponding legend.

* Please ensure that where relevant figures include 95% CIs.

* When a p value is given, please specify the statistical test used to determine it in the legend.

* In the Kaplan-Meier curve(s) please provide the number at risk for each time interval.

* Where data points are discrete, please ensure that they are depicted in the figures as discrete data and not as a continuous line.

* Please provide the unadjusted comparisons as well as the adjusted comparisons in all relevant Tables

* Please specify the variables controlled for in all relevant Tables

* Please move Figure S14 and Table S2 into the main text.

Line 55: Please clarify meaning of performance status

Line 61: Is the date range the date of diagnosis? Please clarify.

Line 62: Please spell out acronyms

Line 65: Please spell out BCLC

Line 70: Please add results using other LLMs (not only for ChatGPT)

Please add results on BCLC-B to the Abstract

Please clarify the version of ChatGPT (and of the other LLMs if relevant) and correct throughout.

Comments from Reviewers:

Reviewer #1: We thank the authors for addressing our previous comments. Some final remarks remain:

1. In the Impact of concordance between LLM-physician on OS section, it is stated that "In BCLC-B, survival was also better in matched patients, although the difference was relatively modest (HR, 1.091; P = 0.008; Fig 2C)". Would a HR > 1.0 indicate poorer survival instead?

2. While it was stated in the reply to Point 5 that "If the observed effects were driven purely by a bias toward patients with better baseline prognosis, we would expect uniformly favorable OS in matched cases across all stages, which was not observed", such a bias would appear to imply more (and more-appropriate) matched cases for the BCLC-A stage (less-severe cases), and fewer (and less-appropriate) matched cases for the BCLC-C stage. It is thus unclear why there would be an expectation of uniformly favourable OS for matched cases across all stages, under the biased baseline prognosis assumption.

However, the details in S15 and S16 Table do suggest that when the LLM makes disagreements, the recommendations are generally similar in expected survival rate to the human assessment. This supports the authors' assertions from the clinical mechanism and within-treatment sensitivity analysis perspectives.

Nevertheless, it may remain more appropriate to analyze clinical utility of the LLM using an "[mean absolute] error in expected survival outcome" metric (which penalizes both under- and over-estimating the survival outcome), rather than a "lower/better expected survival outcome" metric. This is because as there is no counterfactual as to the actual impact of a changed treatment regime (as acknowledged in the response to Point 3 of the previous review), the actual observed survival might more appropriately be considered the ground truth in terms of prognosis correlation.

Reviewer #2: Many thanks for addressing my comments. Just one point is outstanding:

- The Methods section should be explicit about the lack of a formal prompt engineering phase, and detail the background of the individuals that produced the standardised prompts--this is also part of the CHART checklist (point 5).

Reviewer #3: None

[LINK]

---

## [Editor Report · Decision Letter 3]

24 Nov 2025

Dear Dr Han, 

On behalf of my colleagues and the Academic Editor, Amit Singal, I am pleased to inform you that we have agreed to publish your manuscript "Evaluating the Clinical Utility of Large Language Models for Hepatocellular Carcinoma Treatment Recommendations: A Nationwide Retrospective Registry Study" (PMEDICINE-D-25-03475R3) in PLOS Medicine.

PRESS

Sincerely, 

Alison Farrell, Ph.D. 

Senior Editor 

PLOS Medicine